# Strategy Coopetition Explains the Emergence and Transience of In-Context Learning

**Aaditya K. Singh** [1]  **Ted Moskovitz** [2]  **Sara Dragutinović** [3]  **Felix Hill** [4]  **Stephanie C. Y. Chan** [* 4]
**Andrew M. Saxe** [* 1]

## Abstract

In-context learning (ICL) is a powerful ability that emerges in transformer models, enabling them to learn from context without weight updates. Recent work has established emergent ICL as a *transient* phenomenon that can sometimes disappear after long training times. In this work, we sought a mechanistic understanding of these transient dynamics. Firstly, we find that—after the disappearance of ICL—the asymptotic strategy is a remarkable hybrid between in-weights and in-context learning, which we term "context-constrained in-weights learning" (CIWL). CIWL is in competition with ICL, and eventually replaces it as the dominant strategy of the model (thus leading to ICL transience). However, we also find that the two competing strategies actually *share* sub-circuits, which gives rise to cooperative dynamics as well. For example, in our setup, ICL is unable to emerge quickly on its own, and can only be enabled through the simultaneous slow development of asymptotic CIWL. CIWL thus both cooperates *and* competes with ICL, a phenomenon we term "strategy coopetition". We propose a minimal mathematical model that reproduces these key dynamics and interactions. Informed by this model, we were able to identify a setup where ICL is truly emergent and persistent.

## 1. Introduction

Transformer-based large language models (LLMs) show an impressive propensity for in-context learning (ICL)—the ability to use inputs at inference time to adapt behavior and solve tasks not seen in training. ICL contrasts with in-weights learning (IWL), which is standard learning through weight updates. ICL is striking not only for its power, but also because it emerges without being explicitly trained for (as Brown et al. (2020) first observed, when training transformer models on internet-scale language corpora.). Newer work has shown that ICL can sometimes in fact disappear after emerging, i.e. it can be *transient* over the course of training (Singh et al., 2023; Anand et al., 2024; He et al., 2024).

This evolving picture of ICL necessitates a deeper understanding of the dynamics of ICL emergence (and transience). ICL is often viewed to be in *competition* with other strategies (Nguyen & Reddy, 2024; Park et al., 2024), such as IWL, with the tradeoff thought to be modulated by data properties (Chan et al., 2022; 2024), model size (Wei et al., 2023), and/or training time (Singh et al., 2023). While competition may explain why ICL gives way to other strategies through the course of training, the question remains: why does it emerge in the first place (if only to fade away)?

In this work, we aim to extend the *mechanistic* understanding of ICL, which currently focuses on induction heads (Olsson et al., 2022) and their emergence dynamics (Singh et al., 2024), to a richer dynamical setting involving multiple strategies cycling in and out over the course of learning. To do so, we reproduce and investigate the key transience result in a simplified synthetic data setting with a 2-layer attention-only transformer. Using behavioral evaluators, we find the asymptotic strategy after the disappearance of ICL is not pure in-weights learning. Rather, it is a surprising hybrid strategy that we term context-constrained in-weights learning (CIWL, Section 4). The implementation of CIWL takes the form of skip-trigrams (Elhage et al., 2021) distributed across multiple heads in a form of superposition (Elhage et al., 2022). Perhaps even more remarkably, we find that even though CIWL dominates over ICL asymptotically, both strategies *share* critical sub-circuits (Section 5.1), indicating cooperative dynamics between these seemingly competitive mechanisms—a phenomenon we term "strategy coopetition." We borrow the term "coopetition" from game theory, where it describes situations where competitors simultane-

---
[*]Equal contribution  [1]Gatsby Computational Neuroscience Unit, University College London [2]Anthropic AI, work completed while at the Gatsby Unit, UCL [3]University of Oxford [4]Google DeepMind. Correspondence to: Aaditya K. Singh <aaditya.singh.21@ucl.ac.uk>.

*Proceedings of the 42nd International Conference on Machine Learning*, Vancouver, Canada. PMLR 267, 2025. Copyright 2025 by the author(s).

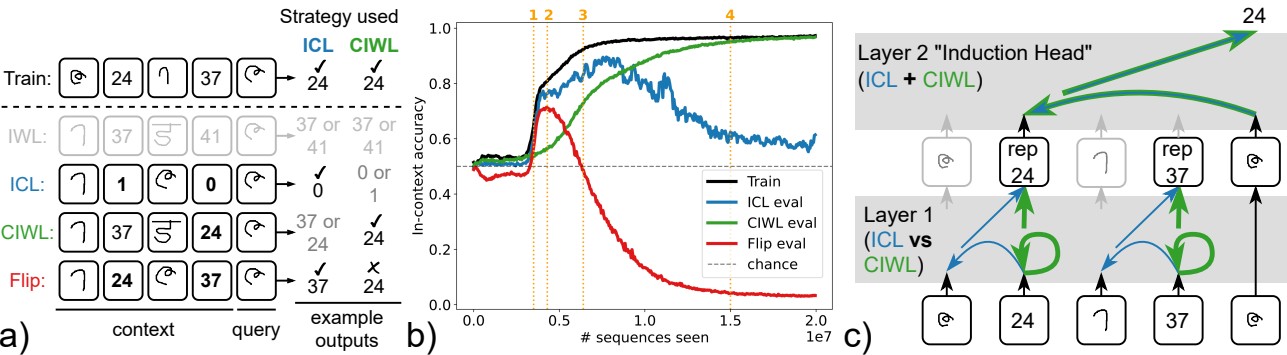

*Figure 1.* **(a)** Example sequences seen during training and evaluation. Training data is "bursty", enabling both in-context and in-weights strategies (the context always contains an exemplar from the same class as the query, but also exemplar-label mappings are fixed throughout training). Evaluation sequences (below dotted line) are designed to measure the presence of different strategies. ICL relies on the exemplar-label mapping in context. IWL depends solely on in-weights information. CIWL requires the correct label in context, but not the query exemplar. The Flip evaluator measures the balance between ICL and CIWL (1.0 means pure ICL, 0.0 means pure CIWL). Bolding indicates OOD exemplar-label pairings. Grayed outputs indicate random selection between the two in-context labels. **(b)** Accuracy on sequences from (a), over the course of training. "In-context accuracy" is computed by restricting the network's outputs to the two labels present in context—this ensures the same chance level (0.5) for all plotted evaluators. ICL transience is clearly visible in blue. IWL is not shown, as we found little-to-no IWL in the networks (Appendix C.1). We annotate four points: 1. the formation of Layer 2 circuits, the canonical "induction head"; 2. ICL strategy dominates network output, as evidenced by peak in the Flip evaluator (red); 3. CIWL strategy matches strength of ICL, as indicated by 50% performance on Flip evaluator; 4. CIWL strategy dominates network output, leading Flip evaluator (red) to be 0 and CIWL evaluator (green) to be 1. **(c)** Illustration of competitive (Layer 1) and cooperative (Layer 2) interactions we find between ICL and CIWL strategies. Both strategies are present in varying amounts through training, as represented by the varying line weights in the Layer 1 circuits: when Layer 1 acts as previous token heads, the network exhibits ICL, but when Layer 1 heads attend to self, the network exhibits CIWL. Crucially, the computation in Layer 2 remains largely unchanged after its initial formation, despite the strategy switch from ICL to CIWL.

ously cooperate and compete with each other.[1] Cooperation enables the emergence of ICL (despite it not being asymptotically preferred), while competition leads to its eventual transience (as previously indicated by Singh et al. (2023)). Notably, ICL emergence can only occur when CIWL is not fully formed (Section 5.4), which may explain the tradeoff seen by Chan et al. (2022): under certain data properties, CIWL forms too "quickly," preventing ICL emergence.

We formalize our intuitions from this case study into a minimal mathematical model capturing coopetition dynamics and explaining the transience behavior. Our model motivates further experiments to modulate the tradeoff between strategies through learning, like reducing the asymptotic bias towards one strategy via data properties. For the case of ICL vs CIWL, we find that matching context and query exemplars removes the asymptotic bias towards CIWL and leads to the persistence of the "faster" ICL strategy.

Our work represents a step forward in understanding how different strategies trade off during learning, through mechanistic investigations on the transience of ICL. We hope our work inspires further work on such coopetitive, dynamical phenomena and enhances intuitions around how capabilities

emerge (and possibly fade) when training transformers.

## 2. Experimental setup

### 2.1. Training details

We train 2-layer attention-only transformers (Vaswani et al., 2017; Elhage et al., 2021) on a synthetic few-shot learning task. We use $d_{model} = 64$, with 8 heads per layer and learned absolute positional embeddings. As is common in mechanistic work (Olsson et al., 2022; Singh et al., 2024), we chose this minimal setting as it sufficed to reproduce key phenomena.[2] We used the Adam optimizer (Kingma & Ba, 2015) with $\beta_1 = 0.9$, $\beta_2 = 0.999$, a learning rate of $10^{-5}$, and a batch size of 32 sequences. All models were trained in JAX (Bradbury et al., 2018). All code is open-sourced at `https://github.com/aadityasingh/icl-dynamics`.

### 2.2. Dataset

Our few-shot learning task consists of sequences of exemplar-label pairs, where image exemplars are drawn from the Omniglot dataset of handwritten characters (Lake et al., 2015). Each character class contains 20 image exem-

---

[1]A classic historical examples is Hollywood studios in the early 20th century, who competed aggressively for talent and audiences, but also collaborated to establish industry-wide standards and jointly negotiate with labor unions.

[2]We still consider various alternative architectures, such as those with MLPs or RoPE (Su et al., 2021), in Appendix B.

plars, and is assigned to a different one-hot label. Images were embedded using a Resnet18 encoder that was pre-trained on ImageNet (He et al., 2015; Russakovsky et al., 2015), before being input to the transformer (Singh et al., 2024). While the original Omniglot dataset has 1623 classes, we follow prior work (Chan et al., 2022) and augment it to 12984 classes by applying flips and rotations. Of these, we use a random 12800 for training. In Appendix B.3, we also considered using different #'s of classes or exemplars, observing similar modulations to Singh et al. (2023) for the duration, timing, and magnitude of the transience effect.

Each training sequence consists of two exemplar-label pairs (the "context"), followed by a "query" exemplar. The model is tasked with outputting the correct label for the query (cross-entropy loss). We train on the *"bursty"* sequences from Chan et al. (2022), where an exemplar-label pair from the same class as the query exemplar is always present in context. This kind of burstiness reflects real-world data distributions like language, and permits both in-weights strategies (since exemplar-label mappings are fixed throughout training) and in-context strategies (since the query exemplar always comes from the same class as one of the two "context" exemplars) (Figure 1a, top row).

### 2.3. Evaluators

To measure in-context and in-weights strategies, we consider four out-of-distribution evaluation sets (Figure 1a).

In the "ICL" evaluator, we replace labels from training with 0 and 1. This invalidates any exemplar-label mappings previously stored in weights. To perform above chance, the model is instead forced to use the mappings provided *in context*, i.e. an *in-context learning (ICL)* strategy.

In the "IWL" evaluator, the query exemplar (and its label) does not appear in the two context pairs, thus preventing the model from using context to perform the task. This forces the model to use knowledge stored in weights, i.e. a pure in-weights learning (IWL) strategy. Performance on this evaluator barely rises above chance (Appendix C.1).

The "CIWL" evaluator is like the IWL evaluator, except that while a matching exemplar to the query is not in context, the correct label *is* in context. Following Singh et al. (2023), the correct label is randomly paired with one of the context exemplars. Like the IWL evaluator, this evaluator can be solved by pure IWL. However, this evaluator also permits a *mixed* strategy where in-context label information can be combined with in-weights information. We refer to the strategy that achieves above chance on this evaluator (but chance on the IWL evaluator) as *context-constrained in-weights learning (CIWL)*. It requires the correct label token in context, but not the full exemplar-label pairing.

Finally, the "Flip" evaluator can be seen as testing for the

model's preference between ICL and CIWL strategies. The two context exemplars have their labels flipped relative to training (e.g. if exemplars X and Y were trained with label mappings X:24 Y:25, the in-context mappings would instead be X:25 Y:24 for this evaluator). The query comes from one of those two classes. If the network prefers the ICL strategy, accuracy on this evaluator would be 1. If the network prefers CIWL, accuracy would be 0, as it would instead output the label that was paired with the query exemplar during training. This evaluator is especially useful in measuring which strategy is *dominant* at each point in training (even when both strategies are present in some form).

## 3. Reproducing the transience of emergent ICL and the convergence to CIWL

Figure 1b shows a reproduction of the key transience phenomena in our simplified setting, with an extended figure in the appendix (Figure 11). ICL emerges and then disappears, as evidenced by above-chance performance on the ICL evaluator (blue). The disappearance of ICL corresponds to a rise in accuracy on the CIWL evaluator (green), indicating that the network is somehow using *just the label information* from context, in combination with some form of in-weights information, to get the right answer.

Notably, there is a significant period of time where ICL and CIWL co-exist (from ∼3e6 to ∼2e7 sequences seen). During this time, the balance between the two strategies shifts from ICL to CIWL, as evidenced by the decrease in the Flip evaluator (red). Early in training, ICL dominates (annot. 2). Then, ICL and CIWL are roughly balanced (annot. 3), before ICL fades and CIWL dominates (annot. 4). This corresponds to a switch in the network's behavior, which first bases its output on the exemplar-label mappings from context, and then eventually to the exemplar-label mappings from training (in combination with the label provided in context).

## 4. The asymptotic "context-constrained in-weights learning" (CIWL) mechanism

Prior work (e.g., Chan et al., 2022; 2024; Nguyen & Reddy, 2024; Anand et al., 2024) has contrasted ICL to "pure" IWL strategies that are completely independent of context. However, we find that the dominant asymptotic mechanism in our networks is dependent on in-weights information but also *context-constrained*, requiring the presence of the correct label in context. In this section, we thus focus on the mechanisms underlying the CIWL strategy, and relegate discussion of auxiliary pure in-weights strategies to Appendix C.1.

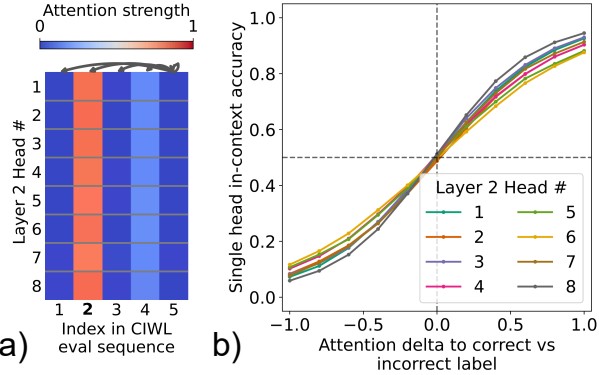

*Figure 2.* CIWL strategy is implemented via skip-trigram-like mechanisms in Layer 2 (L2), with substantial K- and V- composition to Layer 1 (L1). **(a)** Average attention patterns for L2 heads at the end of training. Attention is measured from the query token (index 5) to each token in context. It is computed over CIWL sequences where the correct label is at **index 2** (results for label at index 4 in Fig 12). We see that, at the end of training, L2 heads attend to the correct label, regardless of what exemplar it is paired with in context. **(b)** Task performance as a function of clamped attention delta (difference between post-softmax attention weight) to correct vs. incorrect label token, calculated over 5000 CIWL sequences, when only the given head is active. CIWL accuracy increases as attention to the correct label increases.

## 4.1. Layer 2 heads are skip-trigram-copiers

While label copying might appear to be a separate additive mechanism, we actually find that CIWL is implemented through the use of attention heads in Layer 2 (L2), which act as skip-trigram-copiers (Elhage et al., 2021). This mechanism attends to the correct label, and then copies it forward to the output: "... [label] ... [query] → [label]."

Fig 2a shows attention patterns on the CIWL evaluator. At the end of training, the model correctly attends from the query exemplar to the in-context label that was paired with the query during training (regardless of which exemplar that label currently appears with in context). We also find such trends on in-distribution training sequences and Flip evaluators. Skip-trigram heads only appear in L2, with Layer 1 (L1) heads not contributing directly to the output: ablating L1 → output connections, via the pattern preserving method of Singh et al. (2024), results in a negligible drop in accuracy (98.7% to 98.5%) on the CIWL evaluator.

The second part of this mechanism is the copying forward of the label from context to output. We demonstrate this through the use of causal ablations: we clamp the attention pattern for a given head to give a certain weight to the "correct" label token and the remaining weight to the "incorrect" label token. When considering each L2 head in isolation,[3] we find that modulating this weight is nearly perfectly cor-

related with output predictions (Fig 2b), causally indicating that each L2 head is serving to copy input labels to the output. The combination of attending to the correct label and copying it forward give rise to the CIWL mechanism.

**Layer 2 skip-trigrams may be represented in superposition.** While the averaged results in Fig 2 may indicate that heads are performing similar functions to each other (in contrast to Singh et al. (2024), who found differing strengths for different induction heads), we find more heterogeneity when inspecting responses on individual data points (Fig 15a). It's also surprising that heads do not perfect their attention pattern to the correct label (again differing from the induction heads in Singh et al. (2024)). We suspect that the large number of classes present (12800), compared to the dimension of the model (64) and number of heads (8) lead to the individual skip-trigrams (for each exemplar-label pair) being stored in superposition (similar to sparse features stored in Transformer MLPs (Bricken et al., 2023); preliminary evidence in Appendix C.2).

## 4.2. Layer 1 engages in K- and V- composition, despite not being necessary

While L1 heads do not directly influence network output, ablating them completely (without preserving patterns and values in L2) leads to a big drop in performance. This indicates substantial composition (Elhage et al., 2021) between the two layers (i.e. L2 attention heads are dependent on inputs from L1), despite the fact that skip-trigrams can be implemented with just a single layer: Appendix Fig 13 shows that a 1-layer model can learn CIWL using skip-trigram-like circuits, in line with prior work that describes 1-layer mechanisms for skip-trigrams (Elhage et al., 2021).

To dig into the specific composition between the two layers, we consider various ablations of L1 head outputs on the CIWL evaluator data, while preserving some combination of values, keys, or queries in L2. Recall that accuracy at the end of training on this evaluator is 98.7%, with chance level being 50%. When we preserve everything but values, performance drops to 59.3%, indicating L1 outputs influence L2 values in a crucial way (known as V-composition).[4] When we preserve everything but keys, performance drops to 57.5%, indicating K-composition. When we preserve everything but queries, performance drops to 95.4%, indicating a very small amount of Q-composition (if at all). Thus, the primary reliance of L2 on L1 is in the calculation of L2 *keys and values*, begging the question: Given that a one-layer network could implement skip-trigram-copiers, why does this observed K- and V- composition between layers emerge?

---

[3]This is performed by taking the output of a single L2 head after our clamp, and feeding it through the rest of the network.

[4]As defined by Elhage et al. (2021), the degree of k-, q-, and v-composition is the degree that a previous layer contributes to keys, queries, and values (respectively) of a following layer.

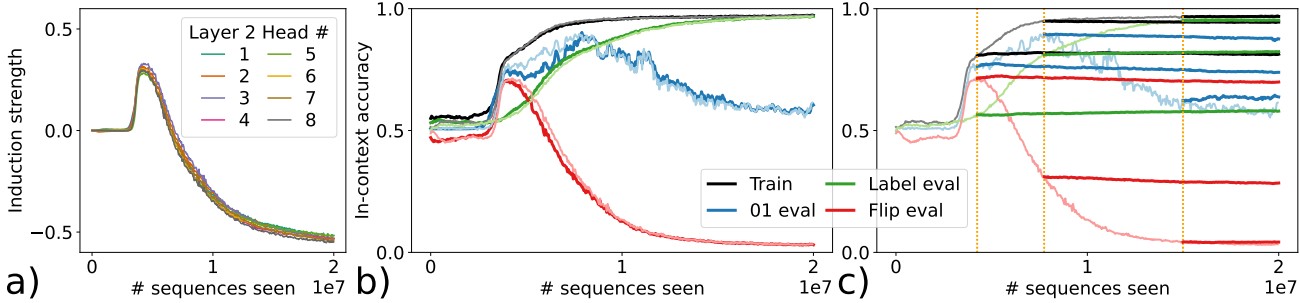

*Figure 3.* The transience of emergent ICL is driven by changes in the function of Layer 1. **(a)** Average induction strength (attention delta to correct ICL token vs incorrect ICL token) on Flip eval data of each Layer 2 head, through the course of training. We see induction circuits emerge then flip, matching the end-of-training attention patterns shown in Fig 2. **(b)** For each available checkpoint, we fix Layer 2 weights to be those from the end of training, and plot performance on each of our evaluators. Using the Layer 2 weights from the end of training (darker curves) reproduces the original behavior (lighter curves; matches Fig 1b) at all points in training after the initial flat portion. This indicates that Layer 2 is not meaningfully changing during the transition from ICL to CIWL. **(c)** For each available checkpoint, we fix the Layer 1 weights to those from a specific checkpoint (marked by the dotted orange vertical lines). After the Layer 1 weights are fixed in this way (darker curves), network behavior doesn't change, as evidenced by the flat lines on all the data we considered.

# 5. ICL emerges, despite not being asymptotic, due to cooperative interactions with CIWL

Induction circuits (which typically underpin ICL strategies, c.f., Olsson et al. (2022)) also show characteristic K- and V- composition, and seem to be present during the transient emergence of ICL in our experiments (Fig 3a). Could the K- and V- composition for the two strategies be related? Do these seemingly competing strategies actually *share* circuit elements? Could that be why ICL emerges, despite not being asymptotically preferred? In this section, we find the answer is, remarkably, yes.

## 5.1. The ICL to CIWL transition is explained by L2 reuse and L1 dynamics

Induction circuits are typically comprised of a "previous token" head in the 1st layer that copies information from the previous token into the next token, and an "induction head" in the 2nd layer that uses the 1st layer to find tokens preceded by the present token. This enables a common form of ICL: "[A*][B*] ... [A] → [B]" (Olsson et al., 2022).

Surprisingly, we find here that the L2 induction heads in the ICL induction circuits are re-used by CIWL with little change. The transition from ICL to CIWL is implemented by a transition in the role of *L1 heads*, which switch from previous-token-attention to attending-to-self (Fig 1c).[5]

This is supported by attention patterns in L1 (Fig 14) and L2 (Fig 2a), and also by two sets of additional experiments, inspired by the notion of progress measures (Nanda et al.,

2023). We take each checkpoint from training, and then fix subsets of the network to the weights from a different checkpoint from the same run. In Fig 3c, we show that fixing the first half of the network (embedding + L1) to that of a checkpoint at a given point (orange dotted lines) leads to very little change in network behavior after that point (flattening of the dark lines as compared to the lighter lines). Conjointly, if we fix the second half of the network (L2 + unembedding) to the checkpoint from the end of training (Fig 3b), we see little difference in behavior after the initial phase change, which indicates that Layer 2 changes are not meaningfully affecting the network after that point.

Notably, this means the canonical "induction heads" (in L2) remain, but they become part of the computational strategy CIWL rather than ICL, due to the change in L1. These results connect to a broader emerging notion that few-shot ICL (Brown et al., 2020) may lie on a spectrum of ICL abilities (Lampinen et al., 2024; Yin & Steinhardt, 2025), which we now show may be connected on the mechanistic level via shared subcircuits.

## 5.2. Learning the ICL strategy is *enhanced* by the availability of asymptotic CIWL

Why does ICL emerge at all, if it is not asymptotically preferred? We find that ICL's emergence may actually be enabled by ICL's shared L2 subcircuit with CIWL. Thus, shared subcircuits may lead to *cooperative interactions*, on top of known competitive interactions (Singh et al., 2023; Park et al., 2024).

To show this, we train networks on "ICL-only" data, where we randomize the exemplar-label mappings across contexts, but keep the mappings consistent within each context. This data can only be solved by ICL, and not by CIWL (since

---

[5]Algorithmically, we believe these heads are essentially learning to map label tokens to prototypical embeddings. It's hard to directly verify such a hypothesis, given the heads' rotation invariance and observed superposition (Appendix C.2), but our evidence for the semantic stationarity of L2 points to such an algorithm.

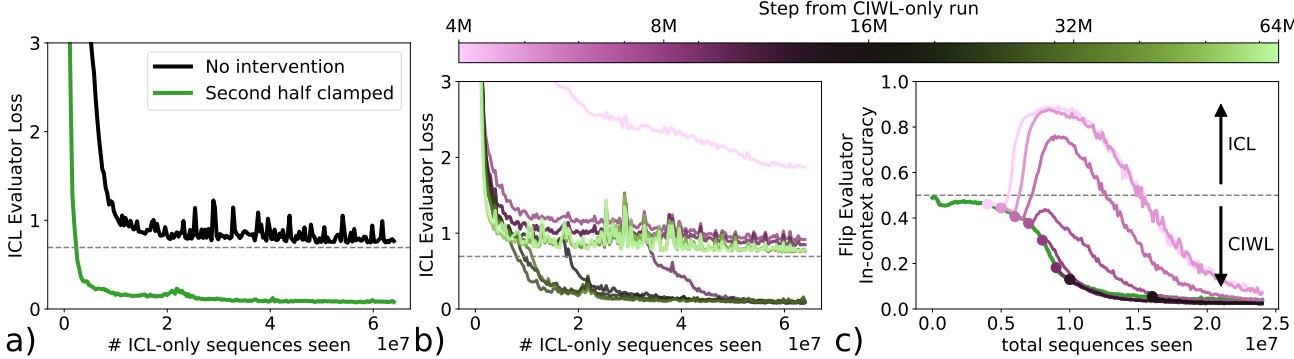

*Figure 4.* ICL emergence is enabled due to *cooperative* interactions with CIWL. **(a)** In this plot, we train networks on "ICL-only" data, i.e. where ICL is a viable strategy but CIWL is not. Without any interventions (black), we hit a loss plateau that greatly slows learning (c.f., Singh et al., 2024). However, if we replace the Layer 2 and unembedding weights with those from end-of-training on our standard training data (green), the network learns quickly. We thus see that these weights, which were part of a CIWL strategy, are reusable for learning ICL. **(b)** We further consider using Layer 2 + unembedding weights from different checkpoints of a "CIWL-only" run, and then training on "ICL-only" data. Early and late checkpoints lead to no learning of ICL, but middling checkpoints (from 9.8M to 31.5M, identified via binary search) do enable ICL learning. **(c)** We continued training different checkpoints from the "CIWL-only" run on our standard training data. Once CIWL has formed (later checkpoints), ICL does not re-emerge even when switching to the bursty data (which otherwise permits ICL).

there are no fixed mappings to learn in weights). [6] When training from scratch, we see the network struggle to learn (Fig 4a, black). Yet when we previously trained on bursty data (where CIWL *was* a viable strategy), we saw ICL emerge quite quickly (Fig 1b). Combined, these form a striking result—ICL actually emerges more easily when it is not the only viable strategy! Presence of an asymptotic CIWL mechanism may be enabling ICL emergence.

To further test this hypothesis, we again trained on ICL-only data, but clamped the weights of the second half of the network (L2 + unembedding), using those from the end of training from our "standard run" on bursty data, after the network has converged to a CIWL strategy. Through this causal intervention on dynamics (using the framework from Singh et al. (2024)), we show that ICL can emerge quite rapidly (Fig 4a, green) with the help of the CIWL weights for Layer 2. These findings connect to the notion that loss plateaus may arise from multiple sub-circuits needing to go right (Singh et al., 2024). When the Layer 2 subcircuits are provided, ICL emerges quickly.

### 5.3. ICL is "close to" the path towards CIWL

One confound in the previous experiment is that CIWL weights at the end-of-training were likely influenced by the earlier presence of ICL (a possible path dependence). Here, we consider a stricter version of the experiment.

We start by training networks on "CIWL-only" data: sequences drawn from the same distribution as the CIWL evaluator, which can be solved by CIWL but not ICL. We

then take checkpoints from different points in training of this CIWL-only run, and use the L2 + unembedding weights to repeat the experiments of Section 5.2; namely, clamped training on ICL-only data. Essentially, we're asking if the L2 weights when learning a CIWL strategy, without influence from ICL, can still enable ICL strategies. Surprisingly, we find that the answer is, for a brief period, yes.

Fig 4b shows training curves of our clamped networks on ICL-only data. Each curve is colored based on the iteration from the CIWL-only run that the clamped weights come from. We see that there is a region from about 10M to 32M sequences seen where ICL can emerge. After this point, L2 weights likely specialize too strongly on a CIWL strategy (as they're being trained on data that only permits such strategies), and thus can no longer be re-used for ICL.

### 5.4. If CIWL is fully formed, ICL will not emerge

To understand why and when ICL emerges, a final critical factor is that CIWL is "slow" to emerge on certain data distributions, relative to ICL. If this were not the case, the competitive interactions might dominate and CIWL might prevent ICL from ever emerging. Indeed, Chan et al. (2022) found many data distributions where in-weights strategies appear quickly and dominate, and in-context strategies do not emerge at all (reproduced in our setup in Appendix B.3).

To show more directly that ICL cannot emerge once CIWL has fully formed, we take various checkpoints from the "CIWL-only" run, and continue training but with bursty data (Fig 4c). When initializing from early "CIWL-only" checkpoints, we see a transient emergence of ICL before CIWL again dominates. At later checkpoints, CIWL persists, with no emergence of ICL. This indicates that ICL could only

---

[6]This resembles the setup used by Singh et al. (2024), except with 12800 classes and 20 exemplars per class.

emerge at the earlier training times because CIWL was not yet fully formed.

## 5.5. Strategy coopetition

Taken together, these results reveal a surprisingly rich interaction between ICL and CIWL: ICL is enabled by CIWL, yet also competing with CIWL for resources in the network. Prior work has pointed to competition between ICL and CIWL, which we now mechanistically observe in Layer 1. In contrast, Layer 2 exhibits cooperative interactions that lead ICL to emerge, despite not being asymptotic. This emergence dynamic has a few requirements:

1. **Useful**: ICL's ability to reduce loss on "bursty data."

2. **On the way**: ICL is "close" to the path of a 2L network learning an asymptotic CIWL strategy.

3. **Fast**: The CIWL strategy emerging "slowly" enough to allow the "faster" ICL to make a transient appearance.

The first factor follows from the dataset design. The second requirement is supported by Sections 5.1-5.3. The third requirement is supported by Section 5.4 and prior work (Chan et al., 2022; Singh et al., 2023; Nguyen & Reddy, 2024).

## 6. A toy model of strategy racing + coopetition

To crystallize these intuitions, we propose a minimal mathematical model that captures the competitive and cooperative interactions at play. Specifically, we consider learning four vectors $\mathbf{a}, \mathbf{b}, \mathbf{c}, \mathbf{d}$ via gradient descent on the following loss function, where $\mathbf{a}^*, \mathbf{b}^*, \mathbf{c}^*, \mathbf{d}^*$ are the true values:

$$\mathcal{L}(\mathbf{a}, \mathbf{b}, \mathbf{c}, \mathbf{d}) =$$

$$\left( \underbrace{||\mathbf{a}^* \otimes \mathbf{b}^* \otimes \mathbf{c}^* - \mathbf{a} \otimes \mathbf{b} \otimes \mathbf{c}||_F^2}_{\text{Mechanism 1 (ICL) Loss}} + \mu_1 \right) \quad (1)$$

$$\times \left( \underbrace{||\mathbf{d}^* \otimes \mathbf{b}^* \otimes \mathbf{c}^* - \mathbf{d} \otimes \mathbf{b} \otimes \mathbf{c}||_F^2}_{\text{Mechanism 2 (CIWL) Loss}} \right) \quad (2)$$

$$+ \alpha \underbrace{||\mathbf{a} \otimes \mathbf{d}||_F^2}_{\text{Competition}}, \quad (3)$$

where $\alpha \geq 0$ is a parameter modulating the strength of competition,[7] and $\mu_1 \geq 0$ modulates the relative asymptotic preference of mechanism 2 over mechanism 1. If $\alpha, \mu_1 > 0$, loss is minimized when $\mathbf{a} = \mathbf{0}, \mathbf{b} = \mathbf{b}^*, \mathbf{c} = \mathbf{c}^*, \mathbf{d} = \mathbf{d}^*$.

This loss function builds on the minimal model of phase change dynamics proposed by Singh et al. (2024), where

---

[7]An alternate interpretation is that $\alpha$ is a fixed version of a Lagrange multiplier for satisfying the constraint that $\mathbf{a}$ or $\mathbf{d}$ is 0.

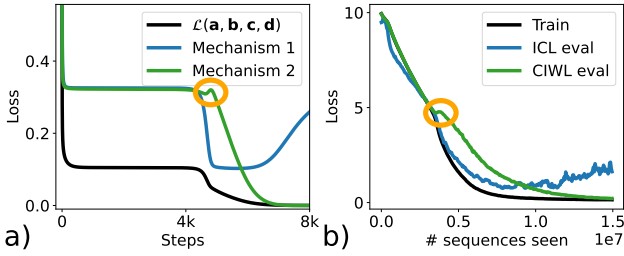

*Figure 5.* Minimal mathematical model captures key phenomena of strategy racing and coopetition. **(a)** Toy model dynamics. **(b)** Loss of real transformer, corresponding to Fig 1b. Notably, both curves exhibit transience behavior in Mechanism 1 (ICL). Intriguingly, the toy model also captures nonmonotonicity in the emergence of Mechanism 2 (CIWL), highlighted in orange.

loss is also computed on tensor products of multiple vectors. The tensor product encodes the intuition that multiple sub-circuits need to simultaneously "go right" in order for the entire circuit to be learned. The model of Singh et al. (2024) is equivalent to the loss for a single mechanism in our model. Our full model has *multiple* circuits (ICL and CIWL) in competition, racing to minimize loss (Section 5.4), but also cooperating (Sections 5.1-5.3) via shared sub-circuits ($\mathbf{b}, \mathbf{c}$).

We model these race dynamics using two mechanisms that interact multiplicatively (lines (1) and (2)). If the offset $\mu_1 = 0$, then minimizing either the loss of Mechanism 1 or 2 will lead to 0 loss. By setting $\mu_1 > 0$, we can enforce an asymptotic preference for one mechanism over the other. Competition is added via the second term (line (3)), which drives $\mathbf{a}$ or $\mathbf{d}$ to 0 in order to achieve 0 loss (if $\alpha > 0$). This corresponds to the network pressure we observe for Layer 1 to be part of an ICL or CIWL strategy. To model cooperative interactions in Layer 2, we re-use vectors $\mathbf{b}, \mathbf{c}$ in both mechanisms. Finally, to model the relative speeds of the two mechanisms, we use different dimensions for $\mathbf{a}$ and $\mathbf{d}$ (larger vectors are slower to learn).

Putting it all together, we simulate the toy model dynamics numerically for the setting of $dim(\mathbf{a}) = dim(\mathbf{b}) = dim(\mathbf{c}) = 20, dim(\mathbf{d}) = 160, \mu_1 = 0.1, \alpha = 0.1$. We show results in Fig 5a, with additional seeds and details in Appendix D. Our simulation shows a transience of Mechanism 1, where its loss first quickly drops with the initial phase change, but then goes back up, indicating that $\mathbf{a}$ is nearly learned but then eventually goes back to $\mathbf{0}$. This return to $\mathbf{0}$ is caused by the asymptotically preferred $\mathbf{d}$ emerging on a slower timescale and the competition term ($\alpha > 0$). These emergence and subsequent transience dynamics mimic those of ICL, with sub-circuits $\mathbf{b}, \mathbf{c}$ representing the shared Layer 2 "induction heads."

Curiously, the toy model dynamics showed a small divot in the formation of Mechanism 2, corresponding to CIWL (highlighted in orange, Fig 5a). Specifically, when Mechanism 1 is nearly fully learned ($\sim 5k$ iterations), there's a

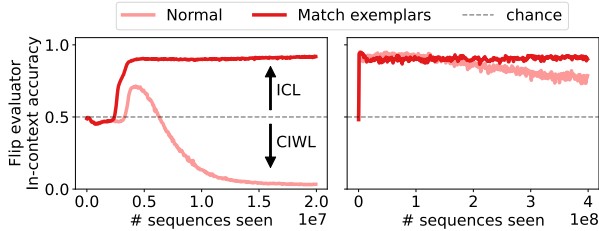

*Figure 6.* Matching context and query exemplars leads to persistent ICL in small (left) and large (right) models, as evidenced by the darker red line converging to accuracy one on the flip evaluator. Larger models are trained with the setup of Singh et al. (2023). The timescale of transience is much longer in larger models (lighter line, right plot), so we don't see a full fading of ICL. Notably, the matched exemplars run shows no indicating of turning downwards.

mild *worsening* of Mechanism 2, before the eventual learning that causes transience of Mechanism 1 and asymptotic dominance of Mechanism 2. When we reexamined the loss curves of our actual transformer, (whose corresponding in-context accuracies are plotted in Fig 1b), we found a similar divot in the loss on CIWL evaluator (Fig 5b)! While we do not fully understand what leads to this brief divot,[8] we believe this mirroring between model and empirics lends credence to this simple mathematical model.

## 7. ICL can be made asymptotically preferred

Equipped with a better understanding of transience, we return to a revised version of the question from (Chan et al., 2022): Are there data properties that incentivize the *asymptotic* emergence of ICL, but don't require it? We tackle this question using intuition from our toy model.

Specifically, if $\mu_1 = 0$, the faster Mechanism 1 (corresponding to ICL) would win asymptotically (Fig 17). Thus, when two strategies are possible, one can be made preferred asymptotically if it's 1) faster and 2) as good asymptotically. We've already established that ICL is faster to emerge than CIWL (Section 5.4) when training on our bursty data, so all that remains is to see if we can find data properties that make ICL "not worse" at the task than CIWL.

The key difference in the mechanisms we find for ICL and CIWL is that CIWL matches query exemplars to *static labels*, while ICL matches to a *variable context exemplar*. This variation may make the "matching" harder for ICL. We ablate this by considering data where the relevant context exemplar is the exact same as the query exemplar.[9] In Fig 6, we find that training on such data makes ICL asymptotic—it

persists after its initial emergence! Furthermore, this finding extends to the larger-scale setup considered by Singh et al. (2023) as well,[10] indicating that the intuitions from the toy model may extend to larger networks.

In combination with Chan et al. (2022), this result would point toward data properties that not only lead to the emergence of ICL—bursty data, with high diversity (# of classes) and variation (# of exemplars) across sequences—but its *persistence* as well: strong correlations in-context—exact matching exemplars.[11] Overall, we found this intervention, motivated by our minimal mathematical model, a compelling illustration of how a better dynamical understanding may lead to data interventions that can control the cycling in and out of strategies throughout training.

## 8. Discussion

When there are multiple ways to solve a problem, when and why does a model "choose" between each strategy?

In this work, we have explored this question by building on previous work demonstrating the *transience* of in-context learning (ICL) in transformers—whereby ICL emerges but then disappears after long training times (Singh et al., 2023; Anand et al., 2024; He et al., 2024, see extended Related Work in Appendix A).

We uncovered a few surprising findings. On a task that was designed to be solvable by both in-weights learning and in-context learning, we find that the model asymptotically prefers an unexpected strategy that is a combination of the two, which we call "context-constrained in-weights learning" (CIWL). After first showing this behaviorally through OOD evaluations, we elucidate the mechanistic implementation of this strategy (Section 4), which may have connections to notions of task recognition (Lin & Lee, 2024) and superposition in larger models (Templeton et al., 2024).[12]

We find that *cooperative* interactions between strategies may lead to the transient emergence of one strategy, even though it is asymptotically dominated by the other strategy. In particular, ICL and CIWL have shared sub-circuitry that enable such cooperation. For the case of transient ICL, we show

---

[8]We speculate that as Mechanism 1 (ICL) forms, the competition forces a brief regression to Mechanism 2 (CIWL), which is slowly learning in parallel.

[9]Note: there are still 20 exemplars per class, making the exemplar-label mapping many-to-one (leaving CIWL the same), it's just that the bursty context exemplar exactly matches the query exemplar (making ICL more effective).

[10]Specifically, 12-layer transformers, with MLPs, trained on bursty sequences of 8 exemplar-label pairs, with the 3 bursty context exemplars matching the query in our intervention.

[11]A possible mechanistic explanation could be that the matching performed by softmax attention cannot be as precise, which leads to an asymptotic preference for CIWL when the bursty exemplars don't exactly match the query.

[12]Specifically, the compression of 12800 class-specific skip-trigrams into 8 attention heads with superposition could motivate research on an analog of sparse autoencoders Bricken et al. (2023) for attention heads (e.g., going from 8 heads to 12800 sparse heads that may individually represent each skip-trigram feature).

that its emergence is due to three key properties:[13] **useful** (ICL helps reduce loss on the task, by construction), **on the way** (ICL is close to the path of a CIWL solution, Section 5.3), and **fast** (ICL only emerges if CIWL is sufficiently unformed, Section 5.4). Eventually, CIWL does dominate and ICL fades due to competitive interactions, as suggested by prior work (Singh et al., 2023). We borrow the term "coopetition" to describe these simultaneously cooperative and competitive strategies.

We crystallize these intuitions from the case study to a more general mathematical model of strategy racing[14] that captures cooperative and competitive interactions. This toy model motivates experiments that identify data properties that do lead to persistent ICL, despite not requiring it, answering some of the original questions of Chan et al. (2022).

The shared sub-circuitry we find may also point to different forms of ICL (e.g., task recognition and task learning) not being as different as previously perceived. While earlier works have argued a tradeoff via competitive interactions (Lin & Lee, 2024; Nguyen & Reddy, 2024; Park et al., 2024), our work suggests a more nuanced take and provides mechanistic evidence for more recent notions that ICL abilities lie on a *spectrum* (Lampinen et al., 2024).

Overall, we hope our work serves to deepen the understanding of ICL and how it may interact with other strategies through the course of training. Beyond ICL, we view our work as a case study demonstrating a few themes that are important to keep in mind, for those of us who wish to understand model behaviors: (1) Models often learn *surprising and counterintuitive* strategies, even for simple tasks. (2) Models are highly *dynamical* and we cannot assume that their strategies remain constant over training, even if they appear stable at a given time. (3) These training dynamics are affected by a kind of *backwards hysteresis*, where later strategies can affect the development of earlier strategies.[15] (4) Alternate strategies are not always strictly in competition and can exhibit *coopetition* by boosting each other.

## Impact Statement

This paper presents work whose goal is to advance the field of Machine Learning. There are many potential societal consequences of our work, none which we feel must be

---

[13]In addition to our "positive" evidence for the found explanation, we reject various alternate hypotheses in Appendix E.

[14]Saxe et al. (2022) also considered the notion of different pathways in a network competing to minimize loss, using the formalism of gated deep linear networks, and found similar intuitions about the "fastest" pathway winning.

[15]We note that this backwards hysteresis complements notions of path dependence, such as the CIWL strategy being influenced by ICL earlier, for which correlates have been found in larger-scale language models (Yin & Steinhardt, 2025).

specifically highlighted here.

## Acknowledgements

We'd like to thank Andrew Lampinen, DJ Strouse, Kira Düsterwald, Jirko Rubruck, Dan Roberts, Basile Confavreaux, Jin Lee, Yedi Zhang, and Murray Shanahan for useful discussions and feedback on early drafts.

A.K.S. and T.M. are funded by the Gatsby Charitable foundation. This work was supported by a Schmidt Science Polymath Award to A.M.S., and the Sainsbury Wellcome Centre Core Grant from Wellcome (219627/Z/19/Z) and the Gatsby Charitable Foundation (GAT3850). A.M.S. is a CIFAR Azrieli Global Scholar in the Learning in Machines & Brains program.

## Dedication

In loving memory of Felix Hill, whose mentorship, vision, and enthusiasm made this entire research program on in-context learning possible (Chan et al. (2022); Singh et al. (2023; 2024), and now this paper). Felix's ideas continue to influence our field and will resonate for many years to come. More importantly, Felix made an immeasurable and lasting impact on the lives of the researchers he mentored and inspired.

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

# A. Extended related work

**Different forms of ICL** Since its first documented emergence at scale (Brown et al., 2020), few-shot ICL in transformers became widely researched. ICL is often contrasted to in-weights learning, where networks learn tasks over the course of training in-weights, rather than adapting to context. However, some (Lin & Lee, 2024) have argued that few-shot learning may be more akin to task recognition, where few-shot prompts serve more to identify a task that the model already knows, rather than learn a new task. This tradeoff may even be modulated by scale (Wei et al., 2023) or training time (Wang et al., 2024). While induction heads (Olsson et al., 2022) may serve as a mechanistic explanation for few-shot task learning, mechanisms for task-recognition-like phenomena have been lacking. In our work, we provide evidence for a context-constrained in-weights mechanism which may help elucidate task-recognition like phenomena by indexing into the label space in context. Furthermore, we hope the shared subcircuitry between different levels of context-usage (CIWL and ICL) may enhance mechanistic understanding of the full spectrum of in-context learning abilities (Lampinen et al., 2024).

**A dynamical understanding of ICL** Since the finding that the emergence of ICL may be a *transient* phenomena (Singh et al. (2023), reproduced by He et al. (2024); Anand et al. (2024) in other settings), many have focused on an enhanced dynamical understanding of ICL. Some have studied emergence dynamics (Reddy, 2023; Singh et al., 2024) of ICL, though only on data that requires ICL to minimize loss. Newer work has considered the tradeoff between strategies throughout the course of training in small (Nguyen & Reddy, 2024; Park et al., 2024) and larger models (Wang et al., 2024), or centered on theoretical modeling (Chan et al., 2024). These works largely focus on *competitive dynamics* between strategies. Our work goes beyond prior work by providing an explanation for the emergence of ICL *when not asymptotically preferred or necessary* due to *cooperative interactions* between different strategies.

# B. Extended behaviors seen in small settings

In this section, we provide additional results on alternative setups we considered—specifically, using other types of positional embeddings, interaction with data properties considered by Chan et al. (2022), and potentially using MLP layers in a 2L network (with absolute positional embeddings). For further exploration of the transience phenomena, we refer readers to the original work of Singh et al. (2023)—our work here focuses on finding a smaller scale setting that allows for study of the key phenomena (with insights that we show may carry to larger settings, Section 7.

All our plots in this section feature accuracy on the IWL evaluator as well. Note that in-context accuracy on this evaluator is meaningless, as the correct label does not appear in context. As a result, we just plot the direct accuracy (for which chance level would be $\frac{1}{\#classes}$). Most of the times, this accuracy stays at chance level, though we note settings where it seems to rise. Further exploration of mechanisms that may be responsible IWL are provided in C.1.

Runs in this section are also often run for longer than our main experiments—we truncated the plots in the main paper to focus on the relevant phenomena. Here, we present the full timecourses we ran in case it's useful for future researchers. All code can be found at `https://github.com/aadityasingh/icl-dynamics`.

## B.1. Different random seeds

To confirm our reproduction, we experimented over a few seeds (for both model initialization and data generation/ordering)—see Figure 7. We found data seed to make little-to-no difference. Model intialization can change the exact timing of transience, but the general profiles of all the curves are the same.

## B.2. Other positional embeddings

We considered absolute sinusoidal positional embeddings (introduced by Vaswani et al. (2017) and used in the original works of Chan et al. (2022) and Singh et al. (2023)) as well as rotary positional embeddings (RoPE, introduced in (Su et al., 2021) and used by (Singh et al., 2024)). However, we found that, for two-layer attention only networks, learned absolute positional embeddings were needed to reproduce the emergence and transience of ICL. While larger networks don't require this added flexibility, we found it necessary to elicit emergent ICL, which we speculate may be due to the difficulty of learning previous token heads in Layer 1 otherwise (Olsson et al., 2022).

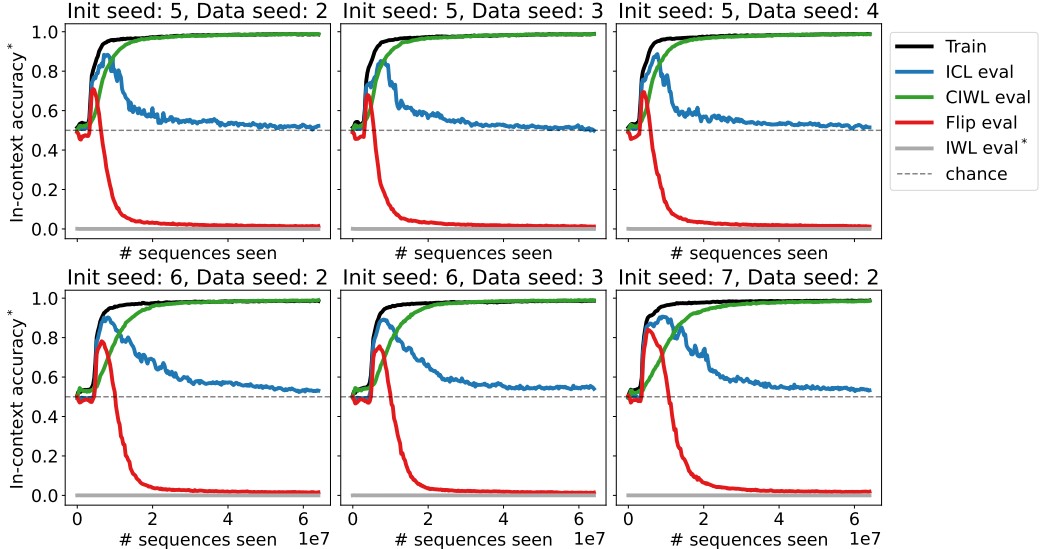

*Figure 7.* Reproduction of our main setup (Section 2) over random seeds. Top left figure is same as Figure 1b.

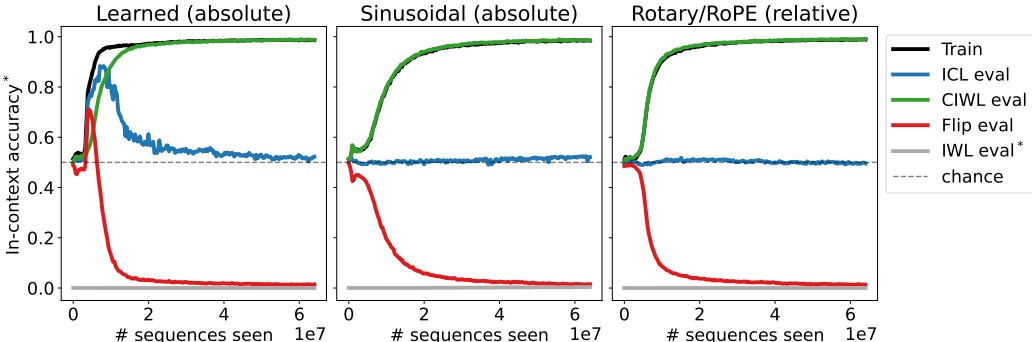

*Figure 8.* Changes in behavior when using different positional embedding schemes. Left figure is the same as Figure 1b. Note that learned absolute positional embeddings are needed to get ICL emergence in a 2L attention-only transformer on our task.

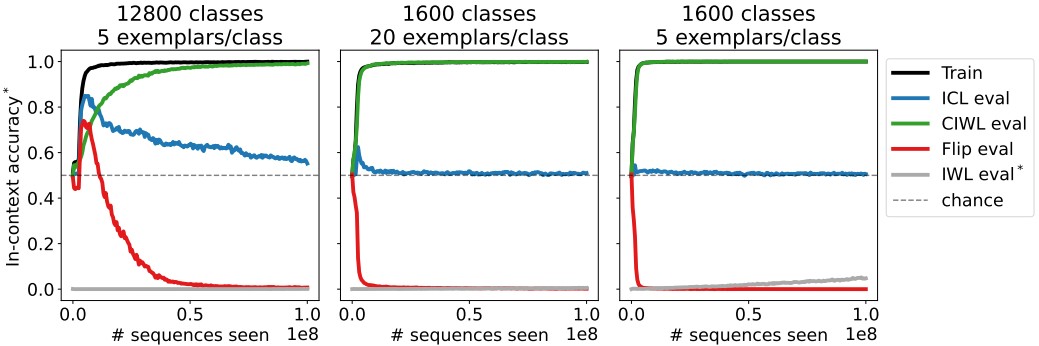

*Figure 9.* Changes in behavior under different data properties, largely reproducing what was seen by Singh et al. (2023).

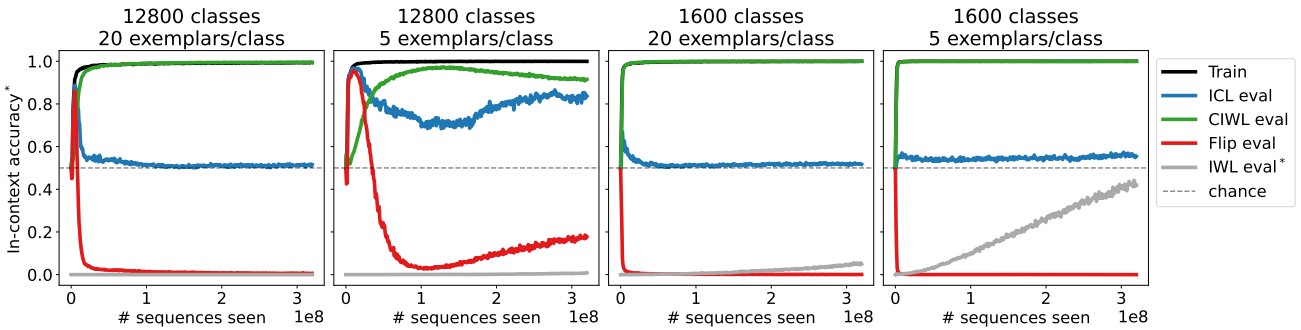

*Figure 10.* Changes in behavior when training 2-layer transformers with MLPs. In our main setting (left), we see similar trends, but when training on 12800 classes with 5 exemplars each, we get transient and resurgent ICL (unlike the corresponding run without MLPs, Figure 9, left) In the rightmost plot, we see significantly increased pure in-weights learning at low numbers of classes and exemplars (as compared to Figure 9), which may be due to the hypothesized role of MLPs in such pathways (Geva et al., 2021; Meng et al., 2023; Singh et al., 2023).

## B.3. ICL does not emerge with lower classes or fewer exemplars

We consider different #'s of classes and different #'s of exemplars. Chan et al. (2022) found that higher numbers of both incentivized ICL emergence, and Singh et al. (2023) found that more classes slowed down transience. In Figure 9, we find that larger number of classes are necessary for ICL to dominate over CIWL for a time (Flip evaluator accuracy $> 0.5$). We still see smaller transience effects with 1600 classes, as long as 20 exemplars are used. When reducing both (1600 classes, 5 exemplars), we see no ICL emergence, and even some asymptotic strengthening of pure in-weights mechanisms.

These findings match what was seen by Singh et al. (2023), and we hypothesize that when # of classes is lower, the CIWL mechanism emerges "too quickly" for ICL to show up (see Section 5.4).

## B.4. Behavior with MLPs

We also considered the use of 2-layer transformers with MLPs as well, as are typically used in practice (Vaswani et al., 2017). We use a standard 4x expansion factor and GeLU activations (Hendrycks & Gimpel, 2023). Results in Figure 10.

While our main setting (12800 classes, 20 exemplars) seems to have similar trends, we observed transience and resurgence in other settings (12800 classes, 5 exemplars), which make it difficulty to claim asymptotic behaviors in these small models with MLP. We thus restricted our main analysis to the attention-only models, as is common for mechanistic work on transformers (Elhage et al., 2021; Olsson et al., 2022; Singh et al., 2024).

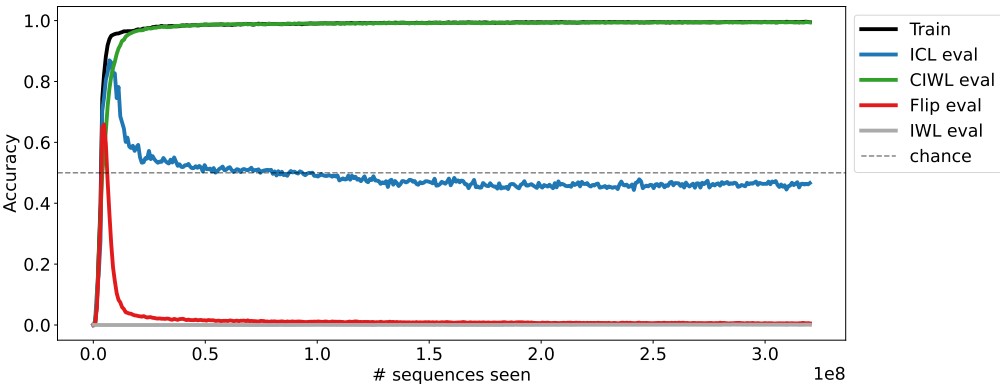

*Figure 11.* Same run as Figure 1b, but run 16x as long. We plot accuracy on the y-axis here (across all 12800 labels) as opposed to in-context accuracy to more compellingly demonstrate saturation of CIWL (green) and minimal learning of IWL (gray). This plot lets us include that networks are in fact not learning a pure IWL mechanism, but rather a context-constrained one which requires the correct label in context.

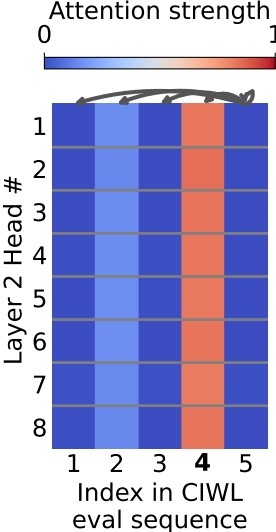

*Figure 12.* Analogous plot to Figure 2a, except the label is inserted at index 4 as opposed to index 2.

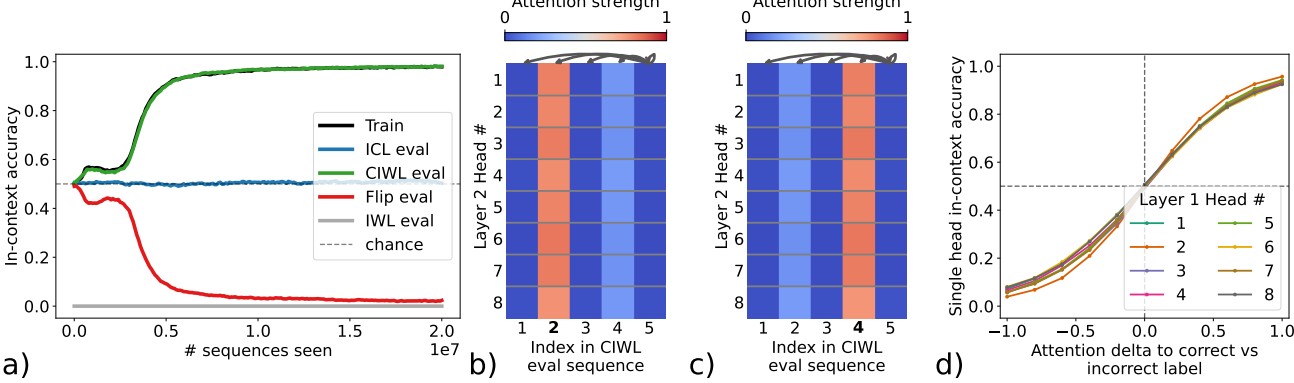

*Figure 13.* A 1-layer transformer trained on our bursty training data can learn skip-trigram-copiers, just as well as the 2-layer model. **(a)** Metrics indicate that the model learns a CIWL strategy. Notably, ICL does not emerge, as we would expect for a 1-layer transformer (Elhage et al., 2021). **(b-c)** Analogous plots to Figure 2a and Figure 12 showing that all heads learn attention from query token (index 5) to the correct token in-context (index 2 or 4 for b, c respectively). **(d)** Analogous plot to Figure 2b showing that all heads are copiers.

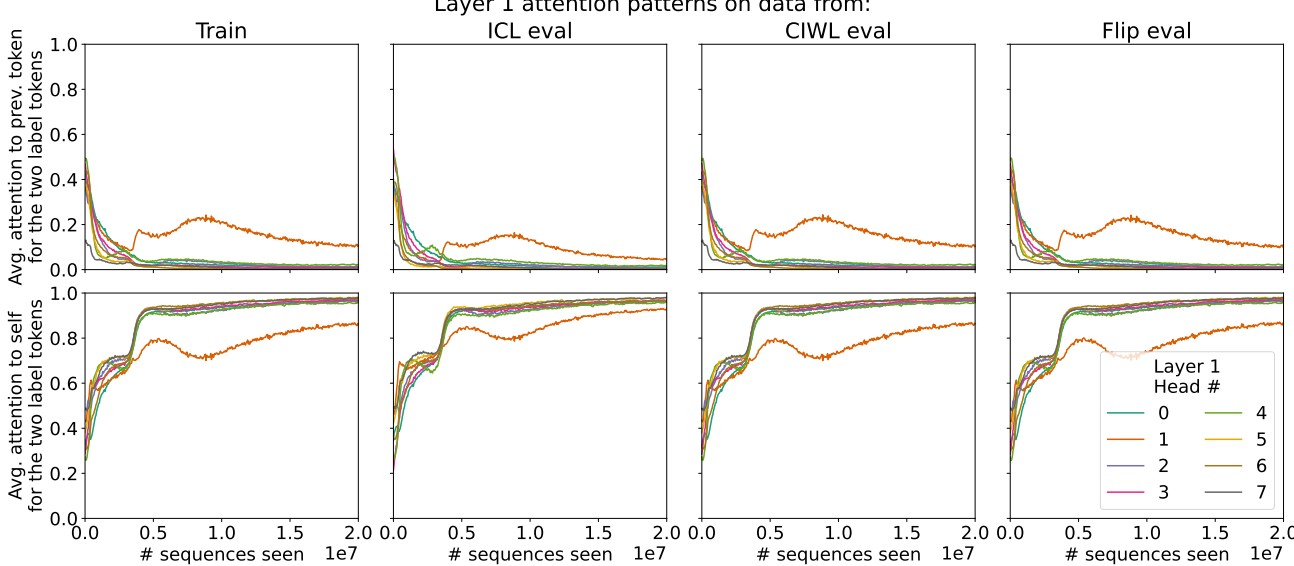

*Figure 14.* Per-Layer-1-head average attention patterns over different data subsets through the course of training. Top row depicts average attention from a label token (at index 2,44) to the previous token (index 1, 3, respectively). Bottom row depicts average attention from a label token (at index 2, 4) to itself. Most of the attention on these tokens is made up of these two pieces (attention to previous and attention to self), though tokens can attend to all previous tokens (standard causal transformer setup). Attention patterns look similar on similar subsets, indicating that Layer 1 heads operate similarly on our out of distribution evaluators. Layer 1 Head 1 appears to have transient previous token behavior, suggesting it may be partly responsible for the emergence of ICL.

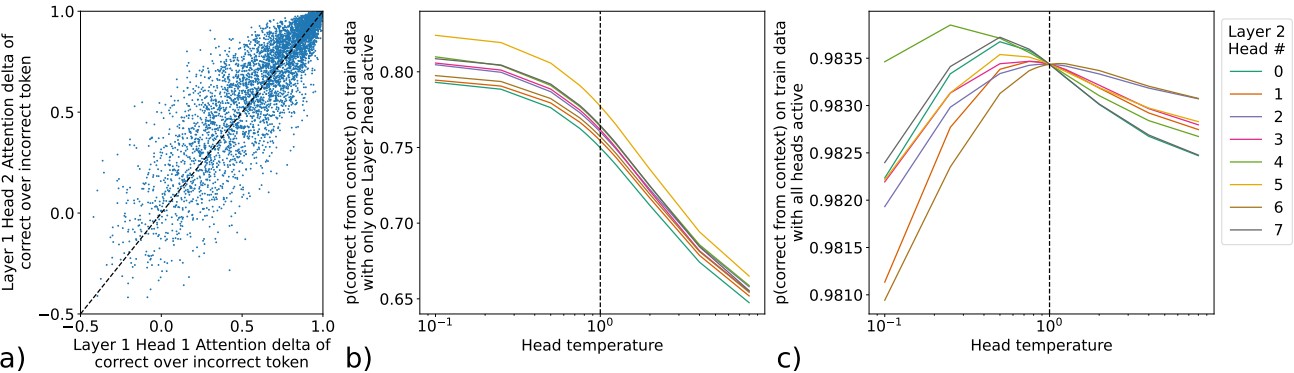

*Figure 15.* Preliminary evidence of superposition. **(a)** While average attention patterns across heads look identical (Figure 2a), patterns on individual points can be quite different (e.g., shown for Head 1 and 2, where each point in the plot represents attention deltas of these two heads for a single evaluation sequence). **(b)** Decreasing the temperature of any Layer 2 head's attention operation, if it is the only Layer 2 head active, increases performance. **(c)** The same temperature decrease does not as consistently affect performance if all heads are active, indicating non-additive interactions.

## C. Additional results

### C.1. Networks barely exhibit *pure* in-weights learning (IWL)

Figure 11 depicts accuracy on a pure IWL evaluator (Figure 1a, gray), which differs from the CIWL evaluator in that the correct label token does not appear in context. This evaluator is the same as that used by Chan et al. (2022); Singh et al. (2023). Like (Singh et al., 2023), we find that networks perform relatively poorly on thie evaluator, instead employing a CIWL strategy (as evidenced by the delta between green and gray curves in Figure 11).

That said, the performance on the IWL evaluator at the end of Figure 11 is 0.08%, which is above chance level (which would be $1/12800 \approx 0.008\%$, indicating that the network may have picked up a tiny amount of pure IWL (or it could be noise). This prompted us to investigate further, and we did uncover a very minor in-weights mechanism.

Specifically, if we ablate all attention heads (so that the network is just the embedding $\rightarrow$ unembedding connection), we find that in-context accuracy on CIWL is 62.46% (compared to 98.74% with no ablation). Note, with this no-heads ablation, the network can't actually attend to any tokens in context, so the "in-context accuracy" can be interpreted as "how likely is the model to output the correct class over a given random class." This value (62.46%) being above chance (50%) indicates the model is able to do this comparison somewhat accurately using just the embedding $\rightarrow$ unembedding pathway.

To make sure this pure IWL mechanism isn't playing a key role in our results, we consider the opposite ablation: zero-out the embedding $\rightarrow$ unembedding pathway, and leave the rest of the network unchanged. We do this by using a pattern-and-value preserving ablation, using two passes through the network. In the first pass, no intervention is used, and patterns and values are cached. In the second pass, we 0 out the embeddings, but use the cached activations for the attention heads, having the intended effect. This ablation leads to 2% drop in accuracy (98.74% to 96.72%), indicating that if the CIWL mechanism, implemented via skip-trigram-copying attention heads, is activate, the embedding $\rightarrow$ unembedding pathway is quite auxiliary.

To conclude, networks do a small amount of *pure* IWL, even when trained on "bursty data," but the contribution of this strategy is negligible compared to the more dominant ICL and CIWL trading off through training.

### C.2. Layer 2 heads may be acting in superposition

In Section 4, we identify that the asymptotic strategy of the network relies on Layer 2 heads acting as skip-trigram-copiers and thus implementing a CIWL strategy. However, we observe some curious features:

1. All heads appear to be doing the same thing on average (Figure 2a).

2. All heads are *imperfect* on average—attention to the correct token never converges to 1 (instead plateauing around 0.8).

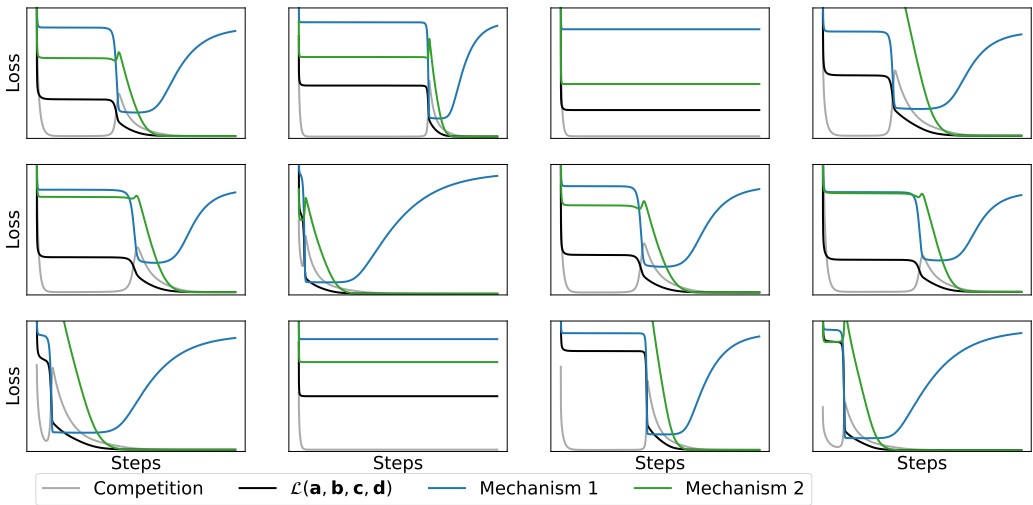

*Figure 16.* Simulations of the toy model over 12 seeds, using the same settings as those for Figure 5: $dim(\mathbf{a}) = dim(\mathbf{b}) = dim(\mathbf{c}) = 20, dim(\mathbf{d}) = 160, \mu_1 = 0.1, \alpha = 0.1$. We additionally show the competition term in gray.

In this section, we lay out preliminary evidence for how these heads may be acting in *superposition*. Our hypothesis was inspired by Elhage et al. (2022), who find that transformer MLP hidden layers may represent sparse features in superposition. The skip trigrams our network is learning are sparse (each sequence only requires comparing 2), and numerous (12800 classes), while the network weights that can be used to represent these are limited (eight 8-dimensional heads).

First, we find that while heads have the same attention patterns on average, their patterns on a given sequence can be quite different (Figure 15a). This would point to some form of distributed computation (which is hidden if only considering average patterns).

The most compelling evidence we find is by considering the interactions between different heads, and their imperfect average attention. If we consider a head in isolation (with all the other heads ablated), we can ask if the imperfection is "optimal." Specifically, we consider artificially decreasing the temperature of the softmax attention on this head,[16] to see if its performance[17] is better or worse. We see in Figure 15b that uniformly decreasing the temperature improves performance, which begs the question: Why doesn't the network learn to do this?

The answer comes from considering how heads interact. If we leave all heads active and then decrease the temperature of a head, we find that overall performance often *goes down*. Each head on its own gets better at solving the task, but the inter-head interactions can often make the network overall worse! These non-additive interactions hint at a tight coupling between heads, and the notion that heads are meaningfully exploiting their dynamic range of attention (instead of the simplified, binary notion of "heads learn to attend to the correct token"). These results are in stark contrast to the additivity of heads found by Singh et al. (2024), and we suggest this is due to a form of superposition.

We conclude by noting that this suggestion is somewhat speculative, backed by preliminary evidence. We hope this heads-in-superposition hypothesis, and preliminary evidence supporting it, can motivate further work on understanding individual attention head function.

---

[16]Note when conducting these experiments, we artificially also only allow the head to attend to label tokens. We know this is what they do in practice, and we wanted to avoid high temperature performance from being artificially hurt by increased attention to non-label tokens.

[17]By performance here, we mean performance on the task if only this head is active. Alternatively, this value can be viewed as a readout of the contribution of this head to the output..

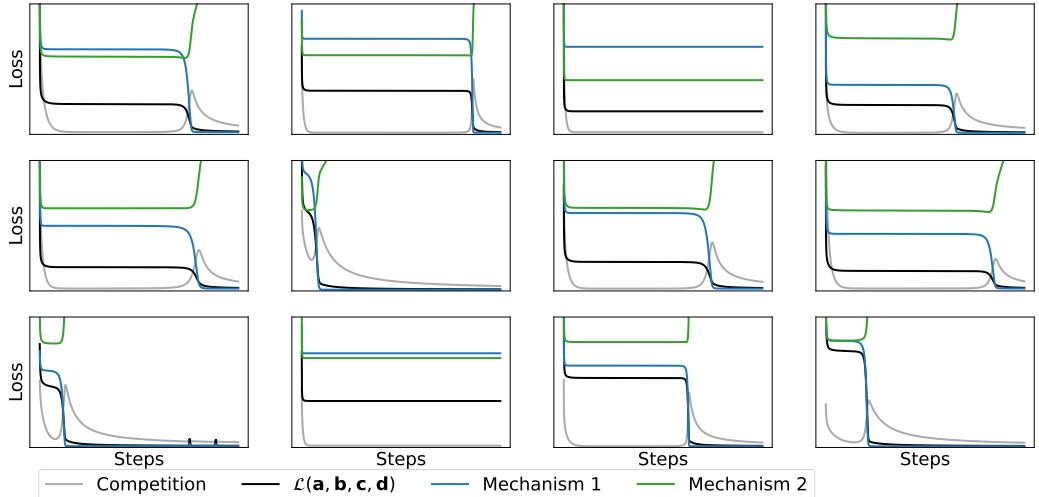

*Figure 17.* Simulations of the toy model over 12 seeds, using the settings: $dim(\mathbf{a}) = dim(\mathbf{b}) = dim(\mathbf{c}) = 20, dim(\mathbf{d}) = 160, \mu_1 = 0, \alpha = 0.1$. Without the asymptotic bias $\mu_1 = 0$, the faster Mechanism 1 gets learned first and dominates. $\mathbf{d} \rightarrow \mathbf{0}$ due to the competition term, which leads to increasing loss of Mechanism 2 (green).

## D. Additional details and seeds on toy model

All vectors in our toy model (Section 6) are initialized randomly with each element being drawn from $\mathcal{N}(0, 1)$. Vectors evolve through gradient descent, with a learning rate of 1. In practice, we use a scaled Frobenius norm in the loss function, where we normalize for the size of the tensors in each expression, to avoid needing to tune the learning rate. Mathematically, this rescaling is isomorphic to a corresponding setting of $\mu_1, \alpha$ and learning rate.

In Figure 16, we show simulations on 12 random seeds (to supplement the single seed shown in Figure 5). While the exact timing in magnitudes of the curves can shift around, the overall dynamical profile (transience of Mechanism 1, the divot in Mechanism 2 formation) is largely preserved. For some seeds, we found that the vectors did not exit the loss plateau (likely caused by a saddle point in the loss landscape, c.f. (Singh et al., 2024)) for the duration of our simulation.

In Figure 17, we show simulations on 12 random seeds with the same settings as Figure 16, except that $\mu_1 = 0$ instead of $\mu_1 = 0.1$. Eliminating the asymptotic bias for Mechanism 2 gets rid of transient behavior, with the faster mechanism (Mechanism 1) emerging and dominating. These intuitions motivated our experiments in Section 7.

All code is open-sourced at `https://github.com/aadityasingh/icl-dynamics`. We're especially excited to see if this toy model can model other types of dynamics (e.g., recent work from Zhang et al. (2025) mentions similar interactions in their discussion).

## E. Rejected hypotheses for the transience of emergent ICL

In the scientific process of figuring out what may cause ICL emergence, given that it's not asymptotically preferred, we considered (and subsequently rejected) many alternative hypotheses. In the spirit of sharing the path used to arrive at a conclusion, we share some of these hypotheses here.

### E.1. Hypothesis 1: Earlier ICL is *necessary* for CIWL to emerge

Given the experiments in the main paper, this hypothesis already seems a bit suspicious—in Figure 13 and Section 5.4 we show that CIWL can be learned without earlier ICL emergence. But what if somehow bursty data requires ICL to emerge first in order to reach a CIWL solution?

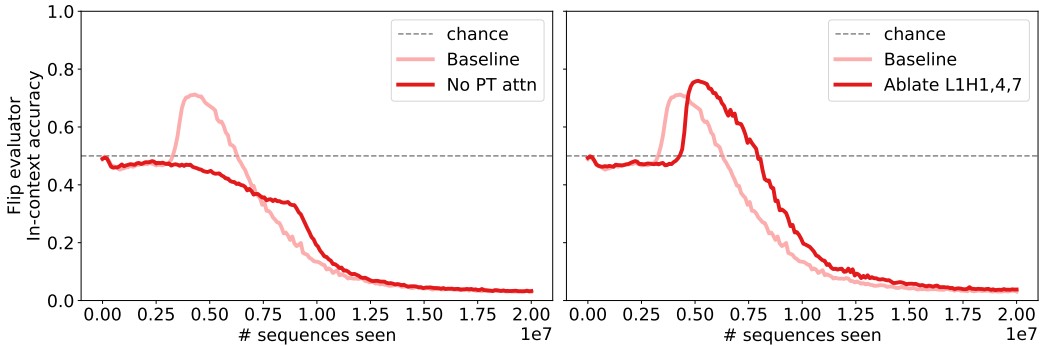

*Figure 18.* Various experiments to reject alternative hypotheses for why ICL emerges, despite not being asymptotic. Recall that accuracy of 1 on the Flip evaluator means ICL is dominant, and accuracy of 0 means CIWL is dominant. **(a)** ICL is not necessary for CIWL emergence. Darker line indicates evolution from a training run where Layer 1 heads were prevented from attending to the previous token, effectively disabling ICL. Lighter line is from standard training (same as Figure 1b). This clamp has relatively little effect on the dynamics of CIWL emergence. **(b)** ICL is likely not an initialization artifact. Darker line indicates evolution from a training run where Layer 1 heads 1, 4, and 7 (which play a crucial role in ICL in standard training) are ablated. While ICL emergence is slightly delayed, it still occurs, and in fact transience is a bit slower as well.

We test (and reject) this hypothesis by considering an experiment where we prevent Layer 1 heads from attending to the previous token, and then training on bursty data. This experiment is a causal intervention on dynamics (Singh et al., 2024), as it prevents ICL emergence but lets other strategies develop naturally. We show results in Figure 18a. Notably, we get similar curves to standard training, with the exception that ICL just doesn't emerge (since it was blocked). Thus, we reject the hypothesis that transient ICL is *necessary* to arrive at asymptotic CIWL.

### E.2. Hypothesis 2: ICL emerges due to a lottery ticket

Another idea may be that ICL simply emerges since typical transformer initializations just happen to be "close" to an ICL solution—so rather than the dynamics leading the model through an ICL solution transiently, it just starts near one and the transience we observe is more of an "initialization effect." Though not quite the same, the notion of lottery tickets (Frankle & Carbin, 2018) is what inspired this hypothesis.

Given our earlier experiments, this hypothesis does feel a bit suspicious as well—for example, we find that ICL transience is quite robust over initialization seeds (Appendix B.1). To offer more evidence against this hypothesis, we again consider a more causal intervention on dynamics (Singh et al., 2024):

If ICL is transient due to initialization effects, this would imply that the circuits being used in ICL are already "largely present" at the start of training. Through ablations on checkpoints from standard training, we identify that (for the main run we analyze throughout the text and presented in Figure 1b) the Layer 1 heads contributing previous-token behavior (Figure 14) to ICL are heads 1, 4, and 7. We establish this by first inspecting attention patterns, and then ablating these heads in checkpoints from around the peak performance of ICL (∼3e6 to ∼1.5e7). Such an ablation on these checkpoints eliminates ICL behavior (as verified by performance on the ICL evaluator dropping to chance level).

We then re-train the model from scratch, with the same initialization, but with these Layer 1 heads clamped off *throughout training*. Presumably, this would eliminate any "already-existing-at-initialization" ICL circuits. Doing so minorly delays ICL emergence (Figure 18b), but does not prevent it. This indicates that their is strong dynamical pressure for ICL to emerge, regardless of initialization, providing evidence against this hypothesis.

