# OpenReview forum: "Strategy Coopetition Explains the Emergence and Transience of In-Context Learning"
_ICML.cc/2025/Conference — ICML 2025 oral_

### Official Review · Reviewer_u2HU · 2025-03-14

**Overall Recommendation:** 5

**Summary:**

This paper systematicallly studies transient dynamics of in-context learning (ICL) in transformers. In particular, the authors identify that after ICL disappears, a hybrid strategy between in-weights and in-context learning called "context-constrained in-weights learning" (CIWL) emerges, which competes with and eventually replaces ICL. Despite this competition, the two strategies share sub-circuits, leading to cooperative dynamics. The paper also proposes a minimal mathematical model to explain these interactions and highlights a setup where ICL remains persistent after long training times.

**Claims And Evidence:**

Yes, the claims made in the submission are supported by clear and convincing evidence.

**Essential References Not Discussed:**

n/a

**Experimental Designs Or Analyses:**

The experimental designs and analyses seem sound/valid to me.

**Methods And Evaluation Criteria:**

Yes, the proposed methods and/or evaluation criteria (e.g., benchmark datasets) make sense for the problem.

**Other Comments Or Suggestions:**

1. Some in-text citations should be done with \citep instead of \cite: line 28-29.

2. The term "attention delta" is used several times in the paper. I can get its meaning from context, I think it would be better to explicitly define the term to avoid confusion.

**Other Strengths And Weaknesses:**

**Strength**

1. The exploration of the transience of ICL is fascinating, and the nuanced findings regarding the existence of CIWL, its interaction with ICL, and its asymptotic dominance provide novel insights into the internal mechanisms of attention-based models and their emergent behaviors. These findings are particularly compelling to me.

2. In addition to the rich results, I appreciate that the authors have developed a simple mathematical model that (1) replicates real-world phenomena and (2) demonstrates some predictive power.

**Weakness**

1. While I appreciate the attempt to develop a mathematical model, I find it somewhat difficult to grasp, particularly in terms of practical interpretation. For instance, the meaning of $\mu_1=0$ is not entirely clear to me—specifically, how it can relate to a characteristic of the dataset (lines 386-392).

**Questions For Authors:**

1. I’m a bit confused about how the mathematical model in Section 6 was developed, as it appears to come up rather abruptly. I understand that there is a pattern match between the model and the ICL experiment described in lines 379-351, but I’m curious about how the specific form of the objective was chosen initially. For instance, why was the tensor product chosen as a component? Why tensor product with three vectors?

**Relation To Broader Scientific Literature:**

The paper built upon prior work such as Singh et al.'23 and Reddy'24, which shows a few new findings related to the transcience of ICL, such as the surprising CIWL and its asymptotic dominance, and its cooperation with ICL.

**Theoretical Claims:**

I didn't check the proofs closely.

---

> ### Author Rebuttal · Authors · 2025-04-01
>
> We thank the reviewer for their review and are glad they found our work "particularly compelling". We've factored in their suggestions and respond to their question below:
>
> The mathematical model presented was largely just our first attempt at extending the "single mechanism" model from Singh et al. (2024) – that's where the tensor product of 3 vectors came from. Our intuition was that, since ICL and CIWL can both "solve the task", we could use the product of the losses as an "OR" operation (a cool symmetry to how the tensor product within a mechanism's loss represents an "AND", as per Singh et al. (2024)). The offset mu1 was added as our results (and the original findings of Singh et al. (2023)) indicated CIWL was always asymptotic. The competition term is between $\mathbf{a}$ and $\mathbf{d}$ since Singh et al. (2024) use $\mathbf{a}$ to correspond to Layer 1.
>
> Further investigation of the toy model would be an interesting avenue for future work – we mostly introduced it to crystallize intuitions and since we were also (pleasantly) surprised by how it captured more nuanced dynamical features (the "divot"). Specifically, we think connecting it to the Neural Race Reduction ideas of Saxe et al. (2022) could be particularly noteworthy.

---

### Official Review · Reviewer_7egA · 2025-03-14

**Overall Recommendation:** 5

**Summary:**

Architecture: 2-layer attention-only transformer (appendix has other models)

Dataset: Omniglot, augmented to 12k+ classes. The majority of the classes are used for training, but the remaining 184 classes are used for testing.

Training sequences are set up with a few shot learning favor and constructed using the exemplars from the training classes. Specifically, each training sequence consists of a context (which contains two exemplar-label pairs) and a query. Training sequences follow the “bursty” structure, requiring that at least one of the exemplars in the context always belongs to the same class as the query.

Testing sequences are constructed using the exemplars from the test classes and they are crafted to evaluate various strategies including ICL, in weight learning (IWL), a new strategy CIWL, and a balance between ICL and CIWL named FLIP.

The main finding is that in the beginning of training, ICL and CIWL cooperate and ICL can emerge. At some point ICL disappears and CIWL comes to the forefront.

## update after rebuttal
I maintain my score and positive impression of the paper.

**Claims And Evidence:**

Yes, the setup is clear and well-justified. The training data design ensures that both ICL and CIWL have an opportunity to emerge, while the test evaluations (e.g., the FLIP test) directly measure whether ICL remains dominant or fades. We see strong empirical evidence for the claim that ICL transience is linked to “coopetition” with CIWL.

**Essential References Not Discussed:**

Not that I’m aware of.

**Experimental Designs Or Analyses:**

Yes, the experimental design is well crafted. The evaluation framework is particularly strong, as it systematically isolates different strategies through controlled test sequences. For example, the FLIP evaluator directly measures whether the model relies on ICL or shifts toward CIWL, providing a clear signal of strategy transition.

**Methods And Evaluation Criteria:**

The main method employed here is really setting up a minimal environment where we can observe “coopetition”. There’s also a portion of the paper dedicated to performing mechanistic interpretability. Here it is discovered that ICL and CIWL share sub-circuits.

**Other Comments Or Suggestions:**

* The term asymptotic is used very frequently throughout the paper. Asymptotic in what is never really spelled out.
* The caption to Figure 1 is very dense. I found it a bit daunting to read in its position. I returned to it after reading far beyond it.
* The x-axis in Figure 1b represents the number of training sequences seen, which implicitly corresponds to training time. Since emergence and disappearance are often framed in terms of training time, making this connection clearer in the text might help readers.
* \\citep should be used in Line 28 and 29 for Olsson and Singh citations
* I’m not sure the footnote on Hollywood studios is a good use of real estate. It’s enough to know that coopetition is a term from game theory and not a gratuitously silly word made up for this paper.

**Other Strengths And Weaknesses:**

This is an excellent example of rigorous deep learning science. The work is compelling and should have broad appeal. A key strength is that it reveals a previously unknown strategy—context-constrained in-weights learning (CIWL)—and its relationship with ICL. The paper exemplifies how well-designed experiments and mechanistic interpretability can lead to novel insights in deep learning.

**Questions For Authors:**

1. How do the authors expect the findings on ICL persistence to generalize beyond Omniglot? Would similar results hold for more complex datasets, such as natural language or vision tasks?
2. In the experiments with deeper networks and MLP layers (Appendix B), did the authors observe any systematic trends in how model depth or capacity influences the timescale of ICL transience? Specifically, does increasing depth delay the transition from ICL to CIWL, or does it accelerate it? If such trends exist, could they be described in terms of scaling laws similar to those seen in other deep learning phenomena?

**Relation To Broader Scientific Literature:**

Recent work has shown ICL to be a transient phenomenon in that it can disappear after long training time. This paper shows that cooperation with CIWL enables the emergence of ICL in the first place, while competition leads to its eventual disappearance and replacement by the CIWL strategy. To the best of my knowledge, the identification of the CIWL strategy is novel.

**Theoretical Claims:**

The paper does not make traditional theoretical claims in the form of formal theorems.

---

> ### Author Rebuttal · Authors · 2025-04-01
>
> We thank the reviewer for their feedback and are happy they think our work is "an excellent example of rigorous deep learning science." We've also noted and updated the paper based on the suggestions, with responses to specific questions below:
>
> > How do the authors expect the findings on ICL persistence to generalize beyond Omniglot? Would similar results hold for more complex datasets, such as natural language or vision tasks?
>
> Given that recent works have found generalization of insights from the Omniglot setting to more naturalistic settings (e.g., language model token embeddings of Singh et al., 2023, and RL setups of Raparthy et al., 2023), we are cautiously optimistic w.r.t generality. For example, recent works on LLMs (https://arxiv.org/abs/2502.14010), point to related phenomena.
>
> > In the experiments with deeper networks and MLP layers (Appendix B), did the authors observe any systematic trends in how model depth or capacity influences the timescale of ICL transience? Specifically, does increasing depth delay the transition from ICL to CIWL, or does it accelerate it? If such trends exist, could they be described in terms of scaling laws similar to those seen in other deep learning phenomena?
>
> We didn't study the timescales as a function of architectural choices, as these were already discussed at scale by Singh et al., 2023 (from reading their paper, the analyses do seem motivated by scaling laws work, though not quantified as exactly, as the reviewer suggests). We did have some earlier experiments related to Layer 1 capacity (e.g., if training with fewer heads active) that showed fewer active heads lead to delayed transience (up to a point – fewer than 3 heads blocked ICL since the network effectively becomes a 1L model). We believe bridging the gap between the rigorous mechanistic understanding of our work, and scaling laws on larger models is an exciting direction for future research.

---

> > ### Comment · Reviewer_7egA · 2025-04-03
> >
> > Thank you to the authors for detailed responses. I maintain my score of 5.

---

### Official Review · Reviewer_Bv3n · 2025-03-25

**Overall Recommendation:** 4

**Summary:**

This paper investigates why in-context learning (ICL), a capability that emerges in transformer models without explicit training, sometimes disappears after extended training periods. The authors study this phenomenon using a simplified experimental setup with 2-layer attention-only transformers trained on a synthetic few-shot learning task based on Omniglot handwritten characters.
The research demonstrates that after ICL disappears, the model does not simply revert to traditional in-weights learning (IWL). Instead, it adopts what the authors term "context-constrained in-weights learning" (CIWL) - a hybrid strategy that requires the correct label to be present in the context but does not need the full exemplar-label pairing that ICL uses. This CIWL strategy is implemented through skip-trigram mechanisms in Layer 2 of the network.

Most notably, the paper uncovers that ICL and CIWL simultaneously compete and cooperate within the model architecture, a dynamic the authors call "strategy coopetition." While the two strategies compete in Layer 1 (where heads switch from attending to previous tokens for ICL to self-attention patterns for CIWL), they share critical subcircuits in Layer 2. This sharing explains why ICL emerges at all, despite not being asymptotically preferred by the model. The authors further develop a minimal mathematical model that reproduces these key dynamics. Their model captures how competition drives ICL's eventual replacement by CIWL, while cooperation enables ICL's initial emergence. Using insights from this model, they identify data conditions where ICL becomes persistent rather than transient - specifically, when context exemplars exactly match query exemplars.

**Claims And Evidence:**

The submission presents several key claims generally well-supported by evidence, but with some major limitations.

The transience of in-context learning (ICL) is convincingly demonstrated in Figure 1b, reproducing previous findings from Singh et al. (2023) and others. The characterization of the asymptotic "context-constrained in-weights learning" (CIWL) strategy is well-supported by:

- Behavioral evidence through specialized evaluators (Figure 1b)
- Mechanistic evidence of skip-trigram-copiers in Layer 2 (Figure 2)
- Ablation studies showing minimal pure in-weights learning (Section C.1)

The core "strategy coopetition" claim is substantiated through experimental interventions such as:

- Figure 3b shows that fixing Layer 2 weights to end-of-training values preserves the overall behavioral trajectory, indicating Layer 2 circuits are shared between strategies
- Figure 3c demonstrates that fixing Layer 1 weights locks in behavior, suggesting competition happens primarily in Layer 1
- Figure 4a-c provides compelling evidence that CIWL enables ICL emergence despite eventually replacing it

The mathematical model in Section 6 reproduces key behavioral patterns observed in the transformers, including the unexpected "dip" in CIWL formation. This strengthens the theoretical understanding of the observed dynamics.

## Problematic claims:
1. The claim that ICL can be made persistent by matching context and query exemplars is supported by Figure 6, though the mechanistic explanation for why this works could be more developed.

2.  The authors briefly discuss the possibility that Layer 2 heads act in "superposition"—that multiple sparse features might share attention heads simultaneously (Appendix C.2). This claim is intriguing and potentially important, yet the current evidence is labeled by the authors as "preliminary." While their exploratory analyses do hint at complex non-additive interactions among heads, this aspect lacks conclusive experimental support. Further investigations—for example, systematically manipulating head temperature or conducting head-specific targeted ablations—would solidify this intriguing hypothesis.

This is concerning  because superposition could significantly impact the authors' interpretation of the mechanisms underlying CIWL, further robust experiments would substantially strengthen this claim. Without more conclusive evidence, this point remains somewhat speculative.

3. The paper claims that its results and explanations have potential implications for larger transformer models and realistic training scenarios (Section 7). However, the presented evidence is mostly limited to synthetic tasks using small-scale, simplified transformer setups. While the authors briefly show preliminary evidence from larger transformer models trained on similar synthetic tasks, they stop short of providing evidence on realistically scaled language modeling tasks or non-synthetic datasets.

While the mechanisms discovered might generalize conceptually, the direct relevance to state-of-the-art transformers trained on natural datasets remains unclear. Explicit evidence from larger-scale empirical studies or tasks closer to practical applications would substantially reinforce this claim.

4. The paper proposes that ICL emerges transiently because it is "close to the path" toward the asymptotic CIWL solution. Although the experiments showing reuse of Layer 2 heads strongly support shared mechanisms between ICL and CIWL, the notion of “closeness” to the path is presented somewhat informally. The authors clearly show that certain intermediate CIWL-only checkpoints allow rapid ICL emergence, but a more explicit or quantitative measure of "closeness" in model space or loss landscape would strengthen this point.

Without a clearly defined notion of "closeness" or detailed visualization of training trajectories (for example, using linear interpolation or functional similarity metrics), the explanation is somewhat abstract. Explicit analysis or visualization of model parameters or activations as training progresses would improve clarity and reinforce this important conceptual claim.

**Essential References Not Discussed:**

Based on my review, the paper generally provides a thorough discussion of relevant literature.

**Experimental Designs Or Analyses:**

I've examined the experimental designs and analyses in this paper and find them generally sound and well-executed. The authors' specialized evaluators (ICL, IWL, CIWL, and Flip) create controlled conditions that effectively isolate specific strategies, allowing clear attribution of model behavior to different learning mechanisms. The Flip evaluator is particularly innovative as it quantifies the relative dominance between strategies rather than just measuring their presence.

The mechanistic analyses provide convincing evidence for the paper's claims. The attention pattern analyses effectively demonstrate the skip-trigram copying mechanisms underlying CIWL, and the authors establish causal relationships through interventions rather than relying solely on correlational evidence. For instance, the attention clamping experiments establish that Layer 2 heads functionally copy label tokens, while the layer-fixing experiments convincingly demonstrate that Layer 2 remains largely static after initial formation while Layer 1 drives strategy changes.

The strategy-specific training experiments offer strong evidence for the coopetition hypothesis. Training on ICL-only data shows difficulty learning, but using CIWL-trained Layer 2 weights enables ICL learning—a key finding that supports the authors' central claim about strategy cooperation. Similarly, the CIWL-only training followed by bursty data effectively demonstrates that ICL emergence depends on CIWL not being fully formed. The paper demonstrates robustness through replication across multiple random seeds, different architectural variants, and various data configurations.

Some aspects that could affect the validity of the analyses include reliance on averaged attention patterns that might mask individual variations in head behavior, the simplified nature of the toy mathematical model compared to actual transformer dynamics, and questions about generalizability from a 2-layer attention-only transformer to more complex architectures. Despite these minor limitations, the experimental designs and analyses provide good support for the paper's claims about strategy coopetition in transformer learning dynamics.

**Methods And Evaluation Criteria:**

The methods and evaluation criteria employed in this paper are well-suited to investigate the transience of in-context learning in transformers. The authors use a simplified architecture (2-layer attention-only transformers) which is methodologically sound for mechanistic studies. This choice follows established practices in transformer interpretability research and allows for clearer attribution of roles to specific components. While this simplification limits generalizability, the authors address this by demonstrating that their key findings extend to larger models (Figure 6, right panel), striking an appropriate balance between interpretability and relevance.
The synthetic few-shot learning task based on Omniglot characters provides a controlled environment where multiple learning strategies are viable, making it an excellent testbed for studying strategy competition. The bursty data design intentionally permits both in-context and in-weights learning, which is crucial for their research questions.

The specialized evaluators (ICL, IWL, CIWL, and Flip) are well-designed for this study. They enable precise measurement of distinct strategies through behavioral signatures:

- The ICL evaluator isolates pure in-context learning by invalidating weight-based exemplar-label mappings
- The CIWL evaluator tests for a specific hybrid strategy requiring context constraints but not full exemplar-label pairing
- The Flip evaluator quantifies the relative dominance between strategies

These behavioral measures are complemented by mechanistic analyses that strengthen the evidence:

- Attention pattern analyses that reveal the underlying computational mechanisms
- Causal ablation studies that establish the functional roles of specific components
- Layer-fixing experiments that isolate the contributions of different model parts

The mathematical model serves as both a theoretical framework and an additional evaluation criterion. By reproducing key behavioral patterns observed in the transformer experiments, it validates the proposed explanation and generates testable predictions.

**Other Comments Or Suggestions:**

- The authors should verify Figure 4c, as the directional arrows for "ICL" and "CIWL" appear to be reversed compared to what's described in the text.
- In Figure 14, there are references to undefined appendices that should be clarified or removed.
- The mathematical notation in Section 6 could be more consistently aligned with the mechanism descriptions in earlier sections to help readers make connections between the empirical findings and theoretical model.
- Finally, some figures (particularly Figures 14 and 15) contain dense information that could be simplified or restructured for clarity.

**Other Strengths And Weaknesses:**

**Originality**

The paper demonstrates novelty in introducing the concept of "strategy coopetition" to explain a previously observed but poorly understood phenomenon of transient in-context learning. By identifying context-constrained in-weights learning (CIWL) as a distinct hybrid strategy and characterizing its mechanistic implementation, the authors provide novel insights beyond what was previously known about ICL transience. The discovery that competing strategies share subcircuits in Layer 2 while competing in Layer 1 represents a conceptual breakthrough in understanding transformer learning dynamics. However, the paper's originality is somewhat constrained by its foundation on existing work on ICL transience and the relatively straightforward extension of previous mathematical models.

**Significance**

This work makes a significant contribution by providing a mechanistic understanding of how transformers transition between learning strategies during training. By explaining why ICL emerges despite not being asymptotically preferred, the authors address a fundamental question about capability emergence in modern AI systems. The finding that ICL can be made persistent through specific data modifications has potential implications for training methodologies, especially if these dynamics extend to larger models as preliminary results suggest. The concept of strategy coopetition could influence how researchers think about capability emergence and circuit formation in neural networks more broadly. However, the significance is somewhat limited by the focus on simplified models and synthetic tasks, raising questions about generalizability to real-world language modeling scenarios. While the authors demonstrate some extension to larger models, more comprehensive validation across diverse architectures would strengthen the work's broader impact.

**Clarity**

The paper presents complex findings with great clarity through well-structured progression from phenomena reproduction to mechanistic understanding to mathematical modeling. The authors effectively use specialized evaluators to isolate and measure different strategies, providing clear operational definitions that facilitate understanding. The visualizations of attention patterns and strategy dynamics effectively communicate key insights, while causal intervention experiments clearly demonstrate functional roles of model components. However, the paper contains dense technical content that assumes substantial familiarity with transformer architecture and mechanistic interpretability techniques. The multiple interrelated experiments and detailed analyses could be challenging for readers to track without careful study. Some important methodological details are relegated to appendices, and the rejection of alternative hypotheses section, while valuable, could be better integrated into the main narrative to strengthen the central claims.

**Questions For Authors:**

1. Your demonstration of strategy coopetition is compelling in 2-layer attention-only transformers, but how confident are you that this mechanism explains ICL transience in larger, more complex models? The preliminary results in Figure 6 suggest some generalization, but what additional evidence or theoretical arguments support the claim that similar dynamics operate in state-of-the-art models?

2. The CIWL strategy you identify bears conceptual similarities to what some researchers call "task recognition" (as opposed to "task learning"). Could you clarify whether you see CIWL as fundamentally the same phenomenon as task recognition, or whether there are important distinctions?

3. Your explanation for ICL persistence when context exemplars match query exemplars is intriguing but somewhat underexplored mechanistically. Have you conducted ablation studies or circuit analyses to understand why this modification equalizes the asymptotic preference between ICL and CIWL?

4. Your toy mathematical model reproduces key dynamics observed in transformers, but how sensitive is this reproduction to parameter settings? Is there a range of parameters for which the model fails to exhibit transience, and what would this tell us about conditions where ICL might naturally persist?

**Relation To Broader Scientific Literature:**

- The identification of "context-constrained in-weights learning" (CIWL) as the asymptotic strategy that replaces ICL builds directly upon Singh et al.'s (2023) discovery of ICL transience, providing a mechanistic explanation for what happens after ICL disappears. CIWL also relates conceptually to Lin and Lee's (2024) work distinguishing between "task recognition" and "task learning" modes of in-context learning. What the authors identify as CIWL shares similarities with the task recognition paradigm, where context serves more to identify what knowledge to retrieve from weights rather than providing new pattern-completion information.

- The paper's core contribution—the "strategy coopetition" framework—extends several research threads. It complements Nguyen and Reddy's (2024) and Park et al.'s (2024) work on competition between strategies, but adds the crucial insight about cooperative interactions between seemingly competitive mechanisms. This reframes prior findings from Chan et al. (2022) regarding how data properties modulate the emergence of different strategies, suggesting that bursty data promotes cooperation between strategies before asymptotic competition takes over.

- The mechanistic analysis showing shared subcircuits between strategies represents a significant advance over previous work. While Olsson et al. (2022) characterized induction heads and their role in ICL, and Elhage et al. (2021) described skip-trigram mechanisms, this paper shows how these circuit motifs can be repurposed between different computational strategies. This relates to Elhage et al.'s (2022) work on superposition in transformers, suggesting that limited model capacity leads to shared computational resources between different capabilities.

- The toy mathematical model extends Singh et al.'s (2024) minimal model of phase changes in transformer learning, incorporating both competitive and cooperative dynamics. It also connects with Saxe et al.'s (2022) theoretical work on the "neural race reduction" and dynamics of abstraction in neural networks, providing further evidence that different learning dynamics can coexist and interact in complex ways during training.

- Finally, the identification of data conditions leading to persistent ICL builds upon Chan et al.'s (2022) investigations of how data properties affect strategy adoption. It also connects to Lampinen et al.'s (2024) recent work on the "broader spectrum of in-context learning," suggesting that different manifestations of context-sensitivity may exist along a continuum rather than as discrete capabilities.

**Theoretical Claims:**

This paper does not present formal mathematical proofs that require verification.

---

> ### Author Rebuttal · Authors · 2025-04-01
>
> We thank the reviewer for their thorough review. We really appreciate your acknowledgement of the strong and thorough evidential support for our main claims.
>
> ## We respond here to the main criticisms:
> > The claim that ICL can be made persistent by matching context and query exemplars is supported by Figure 6, though the mechanistic explanation for why this works could be more developed.
>
> We understand this desire, especially in the context of so many mechanistic explanations for other observed behaviors! However, given the large number of experiments and results, we believe it would be fair to leave this to future work.
>
> > The authors briefly discuss the possibility that Layer 2 heads act in "superposition"... manipulating head temperature... would solidify this intriguing hypothesis.
>
> These results are in Appendix C.2, Figure 15b and c. The overall claim about superposition is meant to be preliminary (and is stated clearly as such). It is not critically related to the main narrative, hence its appearance in the appendix. We chose to include these mentions in the paper to hopefully spur future work in this intriguing direction.
>
> > (Section 7)... Explicit evidence from larger-scale empirical studies or tasks closer to practical applications would substantially reinforce this claim.
>
> Figure 6 shows that our findings scale to other commonly used setups in literature (the 12L transformers and data of Singh et al., 2023, which used the same data as the widely cited Chan et al., 2022). However, we were careful never to claim that such dynamics operate in LLMs – this would need to be left to future work.
>
> As a wider point, there is by now a long tradition of this kind of work on smaller models and simplified setups, which has been well appreciated within the field, and which has indeed shown transfer to larger models and diverse scenarios. We believe it is important to value and undertake this type of work, because it allows us as a field to discover insights which can then be applied and tested in other setups in future work.
>
> ## There are also a few points in the remaining review that indicate that the reviewer may have overlooked some of our analyses:
>
> > Some aspects that could affect the validity of the analyses include reliance on averaged attention patterns that might mask individual variations in head behavior,
>
> We plot individual attention patterns in Figure 15a, in the preliminary evidence for superposition
>
> > The authors should verify Figure 4c, as the directional arrows for "ICL" and "CIWL" appear to be reversed compared to what's described in the text.
>
> As explained on Lines 160-163 when introducing the Flip evaluator, these lines are correct (and simply meant to remind the reader of this indication).
>
> ## Some additional responses:
>
> > In Figure 14, there are references to undefined appendices that should be clarified or removed.
>
> We thank the reviewer for pointing this out (it was simply old text we forgot to edit) and have made the corresponding change.
>
> > The preliminary results in Figure 6 suggest some generalization, but what additional evidence or theoretical arguments support the claim that similar dynamics operate in state-of-the-art models?
>
> We were careful never to claim that such dynamics operate in LLMs, though we do show extension to 12L transformers and common setups from literature (Singh et al., 2023). Our work is meant to build intuitions using rigorous analysis in smaller settings that may inspire work on larger models. For example, https://arxiv.org/abs/2502.14010 points to similar dynamics in larger models (with less mechanistic rigor on dynamics, given the difficulty of such experiments when using LLMs).
>
> > Could you clarify whether you see CIWL as fundamentally the same phenomenon as task recognition, or whether there are important distinctions?
>
> We see them as closely related intuitively, with further mechanistic work in LLMs needed to establish equivalence. Generally, we are wary of overclaiming.
>
> > Have you conducted ablation studies or circuit analyses to understand why this modification equalizes the asymptotic preference between ICL and CIWL?
>
> We believe the "what" has to precede the "why" – our paper mostly focused on the "why" of ICL transience, building intuitions that led us to a setting where ICL is persistent (a new "what"). We believe rigorous investigation of mechanistic explanations here (beyond the intuitions provided in the paper) is beyond the scope of our work, but would be excited for future work to tackle it.
>
> > how sensitive is this reproduction to parameter settings?
>
> We found the model quite robust to parameter settings, with the caveats we mention in the paper (and repeat here for clarity): When mu1=0, the faster strategy would be persistent. When alpha=0, there would be no transience since there's no competition.

---

> > ### Comment · Reviewer_Bv3n · 2025-04-03
> >
> > Dear authors -- thank you for detailed response to my comments. I am satisfied with most of your responses, however, I still recommend aiming to include (or at least give a directional discussion) on the mechanistic explanation of why ICL becomes persistent as it will be helpful for the readers. Overall, I see this paper as making a vital contribution to our understanding the dynamics of in-context learning and I've increased my score to reflect that.

---

> > > ### Author Response · Authors · 2025-04-06
> > >
> > > Thank you for the kind words and updated score -- we will be sure to add a directional, speculative discussion on the mechanistic explanation for ICL.

---

### Decision · Program_Chairs · 2025-05-01

**Decision:**

Accept (oral)

**Comment:**

This paper investigates a phenomenon that has been observed with in transformer models in which in-context learning (the phenomenon by which transformers learn from context without needing to update the weights of the model) can disappear after long periods of training. The authors study this problem  by studying 2-layer attention-only transformers.

Reviewers universally had a positive opinion of this paper and engaged with the authors to clarify certain points that could improve the final version. Nevertheless this paper is a clear accept.